# Large ultrafast-modulated Voigt effect in noncollinear antiferromagnet Mn₃Sn

H. C. Zhao[1], H. Xia [1,2], S. Hu[3], Y. Y. Lv[3], Z. R. Zhao[1], J. He[1], E. Liang[1], G. Ni[1✉], L. Y. Chen[1], X. P. Qiu [3✉], S. M. Zhou[3✉] & H. B. Zhao [1,4✉]

The time-resolved magneto-optical (MO) Voigt effect can be utilized to study the Néel order dynamics in antiferromagnetic (AFM) materials, but it has been limited for collinear AFM spin configuration. Here, we have demonstrated that in Mn₃Sn with an inverse triangular spin structure, the quench of AFM order by ultrafast laser pulses can result in a large Voigt effect modulation. The modulated Voigt angle is significantly larger than the polarization rotation due to the crystal-structure related linear dichroism effect and the modulated MO Kerr angle arising from the ferroic ordering of cluster magnetic octupole. The AFM order quench time shows negligible change with increasing temperature approaching the Néel temperature ($T_N$), in markedly contrast with the pronounced slowing-down demagnetization typically observed in conventional magnetic materials. This atypical behavior can be explained by the influence of weakened Dzyaloshinskii–Moriya interaction rather than the smaller exchange splitting on the diminished AFM order near $T_N$. The temperature-insensitive ultrafast spin manipulation can pave the way for high-speed spintronic devices either working at a wide range of temperature or demanding spin switching near $T_N$.

[1] Key Laboratory of Micro and Nano Photonic Structures (Ministry of Education), and Shanghai Ultra-precision Optical Manufacturing Engineering Research Center, Department of Optical Science and Engineering, Fudan University, Shanghai, China. [2] Department of Physics, Fudan University, Shanghai, China. [3] Shanghai Key Laboratory of Special Artificial Microstructure Materials and Technology and Pohl Institute of Solid State Physics and School of Physics Science and Engineering, Tongji University, Shanghai, China. [4] Shanghai Frontier Base of Intelligent Optoelectronics and Perception, Institute of Optoelectronics, Fudan University, Shanghai, China. ✉email: gni@fudan.edu.cn; xpqiu@tongji.edu.cn; shiming@tongji.edu.cn; hbzhao@fudan.edu.cn

Recent demonstrations of electrical detection and switching of the antiferromagnets offer promising routes to advance the information storage technology of ever-faster speed, higher-density scale, and lower-loss control[1–6]. However, the ultrashort timescale of antiferromagnetic (AFM) spin dynamics makes the electrical detection of the AFM order switching process extremely difficult[7]. The conventional time-resolved magneto-optical Kerr effect (MOKE), which were widely used for probing ferromagnetic (FM) spin dynamics, cannot be used to directly detect the dynamics of AFM order because of the absence of net magnetic moment[6]. Time-resolved X-ray magnetic linear dichroism (XMLD) or diffraction technique adopting large-scale-facility was employed to detect ultrafast AFM spin dynamics[7–9]. Nevertheless, only a handful of such experiments were reported due to the limited accessibility and time resolution. In addition, in a limited number of cases, time-resolved second harmonic generation was used to study the AFM spin dynamics[10,11]. This nonlinear optical technique, however, requires intensive probe laser pulse with resonant wavelength to generate detectable signals.

Recently, Saidl et al. demonstrated a tabletop time-resolved magneto-optical (MO) technique based on the linear optical effect to probe the ultrafast dynamics of the Néel vector in a metallic antiferromagnet of CuMnAs[12]. In this method, the breaking of magnetic equivalence between the spin orientation and that perpendicular to it in the film plane induces the magnetic linear dichroism, which is usually called the Voigt effect. The transient Voigt rotation in the pump probe experiment is given by

$$\Delta P(\Delta t, \theta) = \left(\frac{2Q}{M}\right) \sin 2(\theta - \varphi)\delta M(\Delta t), \quad (1)$$

where $Q$ is the corresponding Voigt coefficient, which scales quadratically with the sublattice magnetization $M$, and $\theta$ and $\varphi$ describe the in-plane projection of spin orientation and light polarization, respectively. $\delta M$ is a pump induced change in the magnetization that depends on the time delay $\Delta t$ between pump and probe pulses. This method was also utilized to probe the Néel order dynamics in an insulating antiferromagnet of CoO[13]. However, these AFM films have in-plane collinear spin configurations, which naturally breaks the magnetic equivalence between the orientations being parallel and perpendicular to the spins to produce the Voigt effect. The magnetic linear birefringence (MLB) was also used to detect the photoinduced magnon oscillations in canted bipartite lattice antiferromagnets with weak ferromagnetism[14–19], where the MLB is related to the collinear AFM order $L$. In antiferromagnet with three sublattices, the noncollinear spin alignment is beyond the description by collinear AFM order. Ultrafast laser induced spin precession dynamics in YMnO$_3$ with triangular AFM spin structure was previously investigated using MLB and/or Faraday effect[20,21], however, its MLB effect is induced by the net magnetic moment related to the precessional spins excited by laser pulses. For such class of noncollinear antiferromagnets with compensated spin alignment, the effectiveness of Voigt effect produced by the inherent noncollinear AFM order in detecting its ultrafast Néel order dynamics remains to be explored.

The hexagonal Mn$_3$X (X=Sn, Ge) systems have recently emerged as a fascinating class of AFM materials and received great interests[22,23]. In such systems, the geometrical frustration in the Kagome lattice of Mn atoms within the (0001) plane leads to non-collinear AFM order with the moments aligned at 120°, forming an inverse triangular spin structure[24–26], as shown in Fig. 1(a). Despite the nearly zero net magnetic moment, the bulk Mn$_3$Sn crystal exhibits surprisingly large anomalous Hall effect as

well as large MOKE owing to its nonzero value of integrated Berry curvature over the occupied bands[22,23]. Its large MOKE response is originated from the ferroic ordering of cluster magnetic octupole formed by the three-sublattice AFM state, rather than from the weak FM moment in the (0001) plane as a result of the competition between the Dzyaloshinskii–Moriya interaction (DMI) and single-ion anisotropy[23,27]. This MOKE response is thus different from the magneto-optical effect originated from the weak ferromagnetism in slightly canted antiferromagnets[28].

The inverse triangular AFM spins in Mn$_3$Sn aligns within the (0001) plane. For Mn$_3$Sn sample with the (0001) plane normal to the sample surface, the in-plane projection of the spins is arranged only in the direction perpendicular to the [0001] axis along the surface. In this case, the in-plane magnetic moments can be viewed as collinear alignment, analogous to the collinear antiferromagnets. Hence, we may expect existence of the Voigt effect in such samples. The Voigt effect, which is quadratic to the sublattice magnetization, has typically dominant role, compared to the MOKE, of inducing polarization rotation of the light reflected from an antiferromagnet[12,29]. In contrast, in ferromagnets, the Voigt effect is usually much smaller than the MOKE[30]. Since the MOKE response in Mn$_3$Sn is not proportional to the net magnetic moment, it is interesting to investigate the Voigt effect and compare its scale with the MOKE response. Furthermore, the certification of the time-resolved Voigt effect for detecting the spin dynamics in such a noncollinear antiferromagnet can promote the understanding of ultrafast dynamics of the noncollinear Néel order which are not easily accessed by other techniques. In particular, it is interesting to investigate the influence of DMI stabilizing the non-collinear spin structure on the ultrafast demagnetization. The demagnetization by ultrafast laser excitation was mainly investigated in magnetic systems with their spin structure dominated by the spin exchange energy and its underlying microscopic mechanism remains debate[31–35].

In this paper, we have observed in Mn$_3$Sn films grown on Al$_2$O$_3$ substrate a large ultrafast-laser-modulated MO Voigt signal, resulted from the inherent noncollinear AFM order. The modulated Voigt signal is at least one order of magnitude larger than the transient MOKE signal. The transient Voigt signal shows a strong temperature dependence from 300 K to the Néel temperature ($T_N$) around 420–430 K[23,36], due to the great diminishment of AFM order near $T_N$. In contrast, the transient reflectivity is nearly temperature independent as a result of the invariable electronic and lattice structure in this range. The transient Voigt rotation angle closely follows a sinusoidal form with a periodicity of 180° when the sample is rotated around its surface normal, whereas the static light polarization as a function of the sample orientation severely deviates from the sinusoidal form. These results reveal that the contribution to the light polarization rotation change by ultrafast laser pulses is much stronger from the modulation of the AFM order than from the hexagonal lattice. Based on the time evolution of transient Voigt signals, we found that the quench time of AFM order shows negligible change with increasing temperature approaching $T_N$. This is in contrast with the pronounced slowing down of the demagnetization processes near the magnetic phase transition in conventional FM and two-sublattice AFM materials. Our results indicate that for magnetic materials with strong DMI to stabilize the noncollinear spin order, the ultrafast spin manipulation can persist near the phase transition temperature.

## Results

**Sample characterization and experimental configuration.** The 40-nm thick Mn$_3$Sn film used in this study was grown on a (1$\bar{1}$02) oriented Al$_2$O$_3$ substrate (see Methods for details). The film

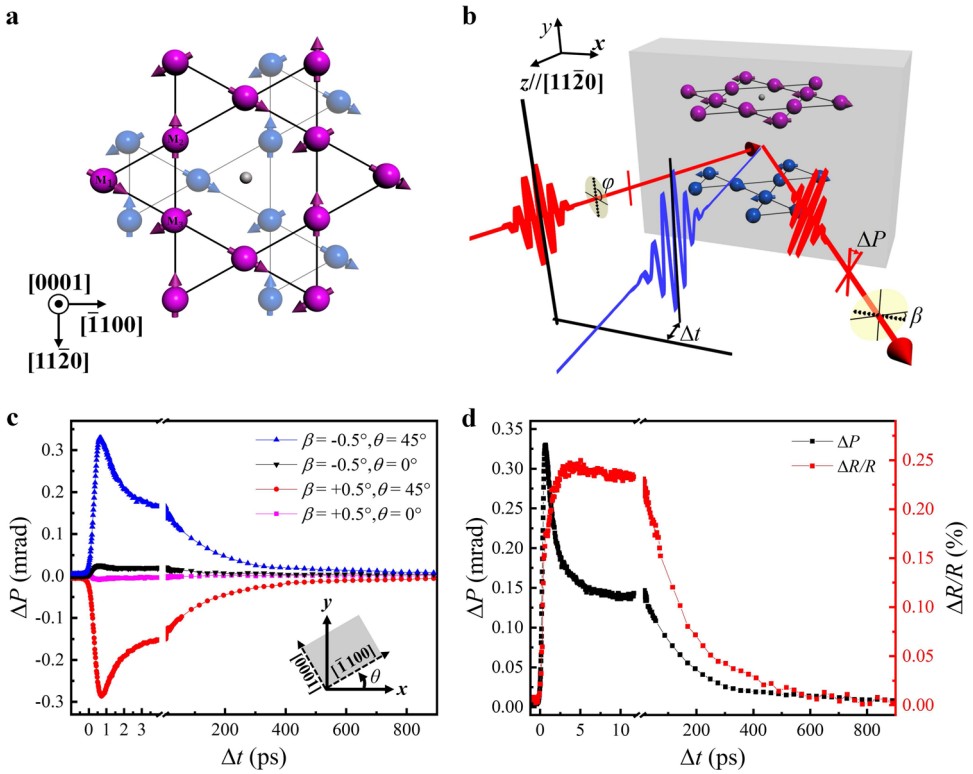

**Fig. 1 Experimental observation of the ultrafast modulated Voigt effect in Mn3Sn film. a** Non-collinear inverse triangular spin structure of Mn₃Sn. Purple and blue spheres with arrows represent Mn spins; small dark gray spheres represent Sn atoms. **b** Schematic experimental setup. The 400-nm pump laser pulse (blue) induces a quench of AFM order, and the 800-nm probe laser pulse (red) detects $\Delta P$ via crossed polarizers. The sample can be rotated around the $z$ axis parallel to [11$\bar{2}$0]. The sample orientation in the schematic is $\theta = 0°$. $\beta$ and $\varphi$ describe the polarizer angle and probe light polarization angle, respectively. $\Delta t$ represents the time delay. **c** Time evolution of the $\Delta P$ at room temperature for sample orientation of $\theta = 45°$ and $\theta = 0°$. $\theta = 0°$ corresponds to the crystallographic direction [$\bar{1}$100] parallel to the $x$ axis. **d** Comparisons of the $\Delta R$ and $\Delta P$ at $T = 300$ K.

surface is normal to the (0001) plane. The X-ray diffraction measurements (see Supplementary Fig. 1) provide pertinent evidence for the high quality epitaxial growth of (11$\bar{2}$0) Mn₃Sn onto the Al₂O₃ substrate. We used an all-optical pump-probe set-up to perform the time-resolved polarization rotation and reflectivity measurements of Mn₃Sn films. Schematic illustration of the set-up is shown in Fig. 1b. The film surface is placed along the vertical $x$-$y$ plane and the surface normal parallel to [11$\bar{2}$0] is defined as $z$ axis. Detailed configuration information can be found in the methods section.

**Time-resolved polarization and reflectivity measurements at room temperature**. Figure 1c shows the time evolution of the polarization rotation change ($\Delta P$) of the reflected probe laser after pump laser excitation (~0.75 mJ/cm²) of the Mn₃Sn film with the (11$\bar{2}$0) surface at room temperature. Here, the dynamic $\Delta P$ curves measured at polarizer angles of $\beta = \pm 0.5°$ from extinction are displayed. For sample orientation of $\theta = 45°$, i.e., the in-plane projection of spin directions intersecting at an angle of 45° with the $s$ polarization of the probe light ($\varphi = 90°$), large $\Delta P$ signals are observed, and as expected, the measured $\Delta P$ signals show opposite signs for $\beta = \pm 0.5°$. The $\Delta P$ rapidly rises up to ~370 μrad within 1 ps and then displays a fast decay followed by a slow recovery process, which will be discussed in detail later. In contrast, for sample orientation of $\theta = 0°$, i.e., the in-plane projection of spin directions being normal to the probe light polarization, only very small transient signals ($\Delta P < 25$ μrad) are detected. These MO signals in our 40-nm thick Mn₃Sn are smaller than the transient Faraday rotation observed in a 300-μm thick FeBO₃ crystal with weak ferromagnetism (>8 mrad) excited at much

higher fluence (30 mJ/cm²) in the transmission geometry[28], but we expect that the MO response of our Mn₃Sn thin film may be several orders larger than the FeBO₃ nanometer-scale thin film considering that the Voigt/MOKE signals measured in the reflection geometry are mainly from the surface of the film.

We note from Fig. 1d that the laser induced reflectivity change ($\Delta R$) displays very different dynamics compared to the $\Delta P$. The $\Delta R$ dynamics exhibit a delayed rising process and also a much slower recovery. These results suggest that the transient polarization change mainly corresponds to the modulation of the Voigt signal, rather than the linear dichroism effect resulted from the hexagonal lattice structure. The Voigt signal arises from the magnetic inequivalence between the in-plane [0001] and [$\bar{1}$100] axes as a result of the inverse triangular spin structure. Here, the in-plane projection of spin directions is collinear with the [$\bar{1}$100] axis, so the Voigt effect, in this case, is analogous to that of the AFM film with an in-plane collinear spin alignment. For $\theta = 45°$, the $s$ polarization can be decomposed into two components with equal amplitude, parallel and perpendicular to the in-plane projection of spin directions. The two components experience different index of refraction, thus leading to the polarization rotation of the reflected probe pulse. For $\theta = 0°$, the $s$ polarization is normal to the spin projection onto the sample plane, thus no Voigt rotation is generated.

We need to point out that the strict description of the MO Voigt effect for non-collinear AFM Mn₃Sn is actually more complicated than simply using the analog of the collinear AFM cubic structure. In Supplementary Note 1, we have analyzed in detail the optical responses of Mn₃Sn, starting from the Maxwell equations, by considering its hexagonal lattice and three magnetic

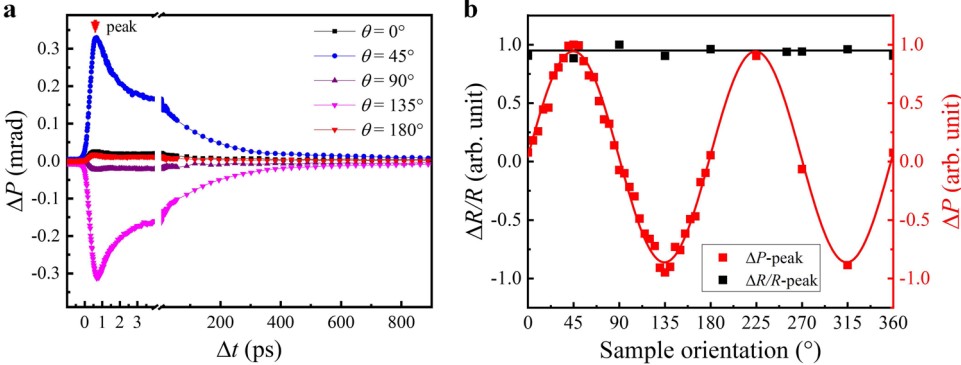

**Fig. 2 Sample-orientation dependence of the transient MO Voigt signals. a** Time evolution of $\Delta P$ for different $\theta$. **b** The peak amplitude of $\Delta P$ and $\Delta R/R$ as a function of $\theta$.

sub-lattices of the inverse triangular spin structure. If neglecting the difference of diagonal components of the permittivity tensor, we obtain the analytical formula identical to Eq. (1) for the Voigt effect. No analytical formula can be derived in the real case of different diagonal components as a result of the hexagonal lattice, however, numerical simulations show that the Voigt signals as a function of the sample orientation $\theta$ still can be well described by Eq. (1), as shown in Supplementary Figs. 2 and 3. Nevertheless, the linear dichroism effect caused by the hexagonal lattice structure leads to the similar orientation dependence of the polarization rotation, and thus it needs to be carefully separated from the modulated MO Voigt signals.

**Sample orientation-dependent results**. To confirm that the $\Delta P$ agrees with the modulation of the Voigt effect, we have further rotated the sample around the surface normal and measured the $\Delta P$ dynamics at various $\theta$. Some typical dynamic $\Delta P$ curves are shown in Fig. 2a, from which we can see that all $\Delta P$ signals reach a peak at the same time delay (~0.6 ps), but their peak amplitudes largely vary for different $\theta$. The peak amplitude as a function of $\theta$, shown in Fig. 2b, closely follows a sinusoidal form with a periodicity of 180°. This result agrees well with the MO Voigt effect in the Mn$_3$Sn film with its (0001) plane normal to the sample surface. For comparison, we also have measured the $\Delta R$ dynamics for different $\theta$ (shown in Supplementary Fig. 4). The overall $\Delta R$ dynamics and peak $\Delta R$ amplitude indicate no $\theta$ dependence, as shown in Fig. 2b.

**Temperature-dependent results**. The AFM spin order in Mn$_3$Sn strongly varies with the ambient temperature, in particularly near its $T_N$[37,38]. In contrast, the lattice structure keeps nearly unchanged in a wide range of temperature from below room temperature to above $T_N$. Hence, we can predict that the transient reflectivity and modulated linear dichroism effect of the lattice should have little dependence on the ambient temperature, whereas the transient Voigt signals must display strong temperature dependence near $T_N$. To examine this prediction, we have performed the temperature-dependent $\Delta P$ and $\Delta R$ measurements. Figure 3a, b shows the dynamic $\Delta P$ and $\Delta R/R$ curves in the temperature range of 300–450 K in the case of $\theta = 45°$. In Fig. 3c, the peak $\Delta P$ and $\Delta R/R$ values extracted from the dynamic curves as a function of temperature are directly compared. It can be clearly seen that $\Delta P$ is diminished with increasing temperature and becomes substantially smaller above 430 K, whereas $\Delta R/R$ almost keep constant. Furthermore, when the sample is rotated to $\theta = 135°$, $\Delta P$ changes signs but exhibits the similarly strong temperature dependence, and in contrast, $\Delta R/R$ is nearly identical to that in the case of $\theta = 45°$, appearing negligible temperature

dependence, as shown in Fig. 3d–f. These temperature dependent results corroborate that the observed $\Delta P$ is mainly resulted from the modulated Voigt effect, rather than from the modulated linear dichroism effect of the lattice structure.

**Field-dependent results**. Although Mn$_3$Sn exhibits very small net magnetic moments at room temperature, a large magneto-optical Kerr effect (MOKE) was reported in the bulk crystal Mn$_3$Sn because of its large nonzero value of integrated Berry curvature resulted from the magnetic octupole structure. When applying the applied field B to reverse the direction of the Mn magnetic moments and thus the ordering of the magnetic octupole, the MOKE signal changes its sign, whereas the Voigt signal should keep constant. To examine the impact of modulated MOKE on the $\Delta P$ signals, we have measured the $\Delta P$ dynamics under applied field from B = −1 T to B = 1 T for $\theta = 45°$ at room temperature, as shown in Fig. 4a. From these dynamic curves, it is not easy to convincingly identify the contribution of modulated MOKE to the $\Delta P$. However, when fixing the time delay $\Delta t$ (~0.6 ps) at the peak position of the dynamic $\Delta P$ curves, and then sweeping B to measure the field dependent $\Delta P$, we can clearly see a hysteresis loop (Fig. 4b). The shape of this dynamic loop is nearly identical to the quasi-static MOKE hysteresis loop measured without the pump laser excitation, as shown in Supplementary Fig. 5. Therefore, this dynamic loop must be due to the transient MOKE response. Here, the slanted MOKE loop may be due to the varied coercive field in different domains of the film. Overall, the coercive field of the MOKE loop is much higher than that observed in the bulk crystal Mn$_3$Sn[23], but actually the relative larger coercive field was typically observed in the Mn$_3$Sn film[34]. Here, the observed MOKE loop is comparable to the hysteresis loop measured by the anomalous Hall effect in the Mn$_3$Sn film[39].

Based on the $\Delta P$ signals shown in Fig. 4, we note that the modulated MOKE rotation angle is more than one order of magnitude smaller than the modulated Voigt rotation angle. This should be due to two factors: (1) For the optical detection using 800 nm s-polarized light, the intrinsic rotation angle induced by the MO Voigt effect is much larger than that resulted from the MOKE, and (2) the quadratic scaling with the magnetization for the Voigt effect leads to the larger modulated $\Delta P$ by ultrafast demagnetization compared to that of the MOKE which was demonstrated to show negligible dependence on the net magnetization[23]. In principle, the magnitude of rotation angle caused by the Voigt effect in the static case can be evaluated by measuring the sample orientation ($\theta$) dependent $\Delta P$ at different temperatures (see Supplementary Note 2). However, we found that the curve of thus measured $\Delta P$ signals is severely deviated from the sinusoidal form expected from the MO Voigt effect or the hexagonal lattice induced linear dichroism effect, as shown in

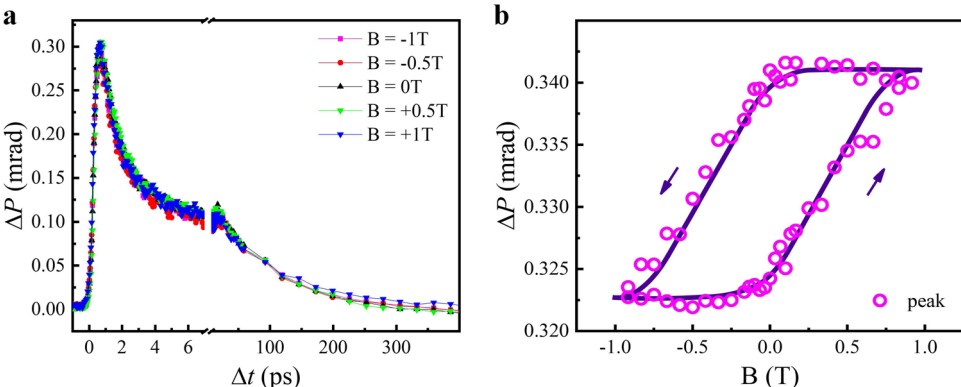

**Fig. 3 Polarization and reflectivity dynamics over a temperature range of 300K-450K.** Temperature dependence of **a** $\Delta P$ versus $\Delta t$ and **b** $\Delta R/R$ versus $\Delta t$ measured by s-polarized probe laser in $Mn_3Sn$ film in the case of $\theta = 45°$. **c** Temperature dependence of the dynamic curves peak of $\Delta P$ and $\Delta R/R$. **d–f** As **a–c**, but in the case of $\theta = 135°$.

**Fig. 4 Modulated MO signals of the $Mn_3Sn$ film for different magnetic field at room temperature. a** Dynamic $\Delta P$ curves at various B. **b** $\Delta P$ loop by sweeping B and fixing the time delay $\Delta t$ at the peak position of the dynamic $\Delta P$ curves. The solid line is a guide to eye.

Supplementary Fig. 6. This deviation is likely due to the nonhomogeneous local lattice strain or defect. As a result, the Voigt effect cannot be reliably separated in the static measurements. Nevertheless, from the comparison of dynamic and static rotation angles as a function of $\theta$ shown in Fig. 2b and Supplementary Fig. 6, one can conclude that the ultrafast laser induced contribution from the lattice to the $\Delta P$ is much smaller than the contribution from the modulated Voigt effect. This actually explains why the $\Delta P$ signals are greatly diminished around the $T_{\mathrm{N}}$.

## Discussion

In order to have deep insights into the AFM spin dynamics impinged on the transient $\Delta P$ signals and compare it to the lattice dynamics depicted by the transient $\Delta R$ signals, we adopt the following convolution function to fit the curves of $\Delta P$ versus $\Delta t$

$$\Delta P(\Delta t) = A_1 \left(1 - e^{-\frac{\Delta t}{\tau_1}}\right)' \otimes \left(A_2 e^{-\frac{\Delta t}{\tau_2}} + A_3 e^{-\frac{\Delta t}{\tau_3}}\right). \quad (2)$$

Here, $A_1$ represents the amplitude of the rising component, $A_2$ and $A_3$ denote the relative amplitude of the two recovery processes with their sum equal to one, and $\tau_1$, $\tau_2$, and $\tau_3$ denote the rising and decaying time constants. Figure 5a, b shows typical fitting curves of $\Delta P$ versus $\Delta t$ using Eq. (2) at different temperatures. The fitting results are in good agreement with the measured data. Basically, the dynamics of $\Delta P$ can be divided into three stages, i.e., a very fast rising process (stage I), and a relatively fast recovery process and a slow one (stages II and III).

The stage I corresponds to the ultrafast quench of the AFM spin order. We obtain from the fitting the quench time $\tau_1 \approx 230$ fs for $T = 300$ K. Such a demagnetization time scale is typically found in many FM metals or AFM oxides[40,41], and it is ascribed to the efficient spin thermalization as a result of strong electron spin interactions ($\tau_{es}$). However, in contrast to the significant slowing down of demagnetization time for $T$ approaching the magnetic phase transition in conventional FM metals and two-sublattice AFM oxides[13,42–44], the quench time in Mn$_3$Sn shows negligible change near $T_{\mathrm{N}}$, as illustrated for the $\Delta P$ dynamics shown in Fig. 5b, from which we obtain $\tau_1 \approx 270$ fs at $T=430$ K. More evidences of the temperature-insensitive quench time are shown in Supplementary Figs. 7 and 8, which show the fitting of $\Delta P$ dynamics at $T = 410$ K and demagnetization time $\tau_1$ at $T = 300$–$450$ K, respectively. Moreover, we find that the fast recovery time of AFM order in Mn$_3$Sn also displays little variation with $\tau_2 \approx 1$ ps for different $T$. This fast recovery is likely due to the fast energy transfer from the spin system to the lattice ($\tau_{sp}$), as evidenced in the $\Delta R$ dynamics shown in Fig. 5c, d.

Similar to the $\Delta P$ dynamics, the $\Delta R$ dynamics can also be divided into three stages. However, two of them are the rising processes, and only one recovery process is involved. We use the power function $\Delta R(\Delta t) = \left[A_1'\left(1 - e^{-\frac{\Delta t}{\tau_1'}}\right) + A_2'\left(1 - e^{-\frac{\Delta t}{\tau_2'}}\right)\right] \cdot e^{-\frac{\Delta t}{\tau_3'}}$ to fit the curves of $\Delta R$ versus $\Delta t$, and obtain the fitting results in good agreement with the measured $\Delta R$. Apart from the fast rising process at stage I possessing a time scale of $\tau_1' \sim 600$ fs due

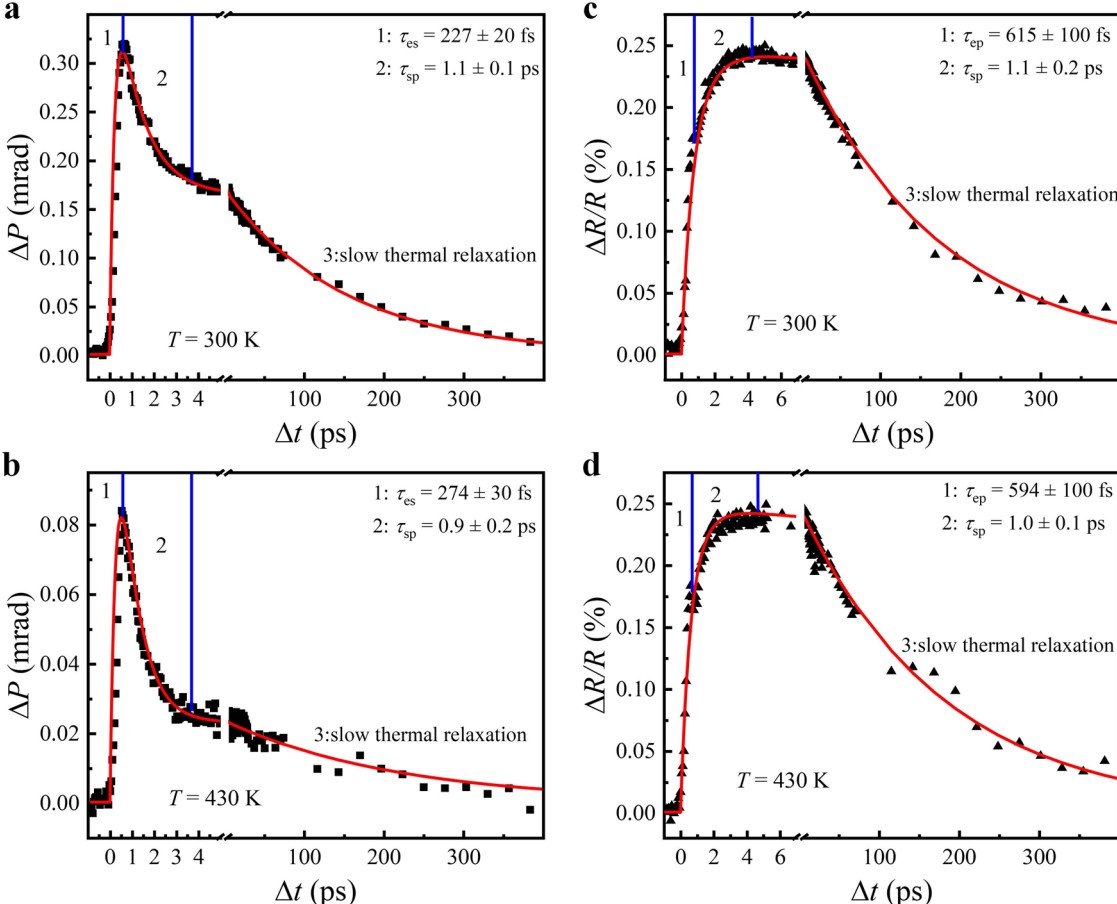

**Fig. 5 Simualtion of pump-induced demagnetization and reflection.** Dynamic $\Delta P$ signals at 300 K (**a**) and 430 K (**b**), and dynamic $\Delta R$ signals at 300 K (**c**) and 430 K (**d**). The solid curves correspond to the fitting results. $\tau_{es}$, $\tau_{ep}$, and $\tau_{sp}$ represent the energy transfer time in electronic-spin, electronic-lattice and spin-lattice system, respectively.

to the electron lattice interaction ($\tau_{ep}$), we note that the rising process at stage II has a characteristic time constant of $\tau'_2 \sim 1$ ps, which coincides with the time of fast recovery process of the AFM order. Hence, this extra rising $\Delta R$ is just because of the further energy transfer from the spin system to the lattice ($\tau_{sp}$).

As we have pointed out above, in FM transition metals[44], chalcogenides[43], and oxides[42], the ultrafast demagnetization shows pronounced slowing-down behavior near the phase transition $T_C$. This behavior occurs because of the weaker electron–spin coupling or smaller exchange splitting[44]. (i) For weak electron–spin coupling, the spin system is heated slowly. After electron–phonon equilibration, the spin system is still relatively "cool" with respect to the electron and phonon systems, and demagnetization proceeds for a longer time. (ii) For smaller exchange splitting energy $\Delta_{ex}$ when $T$ approaching $T_C$, the rising of spin temperature proceeds much slower since the rate of demagnetization depends on $\Delta_{ex}$ as proved in the microscopic three-temperature model (M3TM)[45].

In non-collinear AFM $Mn_3Sn$, the demagnetization time is almost invariant, which can be explained from the following two aspects. (i) The coupling between the electron and spin systems in $Mn_3Sn$ is still strong when $T$ approaches $T_N$, which maintains the fast rise of spin temperature after laser excitation. This stable coupling is also manifested from the energy transfer from the spin system to the lattice, where the transfer time has negligible change in the temperature range from 300 K to near $T_N$ (shown in the $\Delta R$ dynamics in Fig. 5c, d). (ii) For hexagonal $Mn_3Sn$, the stable spin structure is determined by minimizing the Hamiltonian of system, including anisotropy energy, exchange energy, and DMI energy[24]. Especially, the DMI defines and stabilizes the inverse triangular spin configuration[46,47]. When the temperature is close to $T_N$, the diminishment of the inverse triangular spin order is mainly due to the decreasing DMI with thermal disorder[48,49]. In contrast, the exchange energy has little effect on the spin structure, and hence the rising rate of spin temperature and the consequent demagnetization time are barely affected.

In conclusion, it is found that in $Mn_3Sn$ films with hexagonal lattice and inverse triangular spin structure, the reflected light polarization change induced by the ultrafast laser modulated MO Voigt effect is much stronger than that caused by the strain- and crystal-structure-related modulation, thus allowing the study of noncollinear AFM spin dynamics using the MO Voigt effect. We also have observed that the modulated MO Voigt angle is more than one order of magnitude larger than the transient MO Kerr angle. Although the modulation of AFM spin order in $Mn_3Sn$ occurs at the time scale similar to the typical demagnetization time in the conventional magnetic metals and oxides, it displays weak temperature dependence even near $T_N$, in markedly contrast to the pronounced slowing down of the demagnetization for the $T$ approaching $T_C$ in the latter materials. This contrastive behavior provides further insights into the dominant role of the exchange splitting on the ultrafast demagnetization. For magnetic materials with strong DMI to stabilize the noncollinear spin order, the spin manipulation by the ultrafast laser pulse near the magnetic phase transition temperature may be as fast as that at low temperatures. This work reveals an easily accessible avenue to study ultrafast Néel order dynamics in noncollinear antiferromagnets, and the finding of temperature-insensitive ultrafast spin manipulation has significant implications for the application of high speed spintronic devices either working in a wide range of temperature or demanding the switching of magnetic order at elevated temperature near $T_N$.

## Methods
**Sample preparation**. The 40-nm thick $Mn_3Sn$ film were grown using magnetron sputtering system with a base pressure smaller than $1.0 \times 10^{-7}$ Torr. The $Al_2O_3$ substrate temperature was set at 375 °C with the heating rate of 1 °C/s. Sample deposition was started after 20 min when the temperature reaches 375 °C. The Ar gas pressure and sputtering DC power for $Mn_3Sn$ were 2.0 mTorr and 40 W, respectively. The film was post-annealed at 375 °C for 1 h following the film deposition in vacuum to promote crystallization. We thus obtained the film with its surface parallel to the (11$\bar{2}$0) plane, and the (0001) plane normal to the surface.

**Time-resolved polarization rotation and reflectivity measurements**. The time-resolved polarization rotation and reflectivity measurements of $Mn_3Sn$ films were performed by using an ultrafast all-optical pump-probe set-up. The 800-nm probe laser pulses were delivered from a Ti/sapphire amplifier laser system producing 100-fs pulses with a repetition rate of 1 kHz. The 400-nm pump laser pulses were generated by frequency doubling the 800-nm pulses in a beta barium borate (BBO) crystal. The pump laser was nearly normal incident on the sample at the fluence of ~0.75 mJ/cm² for the measured data except special note. We tuned the pump laser polarization with a zero-order wave plate. The s-polarized probe laser with a smaller fluence was directed onto the sample at an incident angle of 40° in the horizontal x-z plane. We used the oblique incidence to measure the transient Voigt effect and the longitudinal MOKE for direct comparison. The polarization rotation measurements by rotating sample in this configuration can also help to exclude the contribution of net magnetization. To detect the polarization rotation change ($\Delta P$) of the reflected probe pulses, we adopted the crossed polarizers with the analyzer setting at a small angle ($\beta$) with respect to the extinction position. The transient reflectivity ($\Delta R$) was measured without inserting the analyzer. For room temperature measurements, the sample was placed on a motorized rotation stage. For temperature dependent measurements from 300 K to 450 K, the sample was placed in a cryostat to prevent oxidation. A magnetic field up to 1 T was applied within the sample plane along the x axis.

## Data availability
The sample information data used in this study are provided in the Supplementary Information and Method.

## Code availability
The code that supports the numerical analysis for the Voigt effect is available from the corresponding author upon reasonable request.

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

## Acknowledgements

Authors gratefully acknowledge the financial support from National Key Research and Development Program of China (2016YFA0300703), National Natural Science Foundation of China (11774064, 51971064).

## Author contributions
H.B.Z., S.Z. and X.Q. conceived this project. S.H., Y.L. and S.Z. grew the sample and performed the characterization. H.C.Z., H.X. and Z.Z. built the pump-probe system, performed the measurements, and analyzed the data. H.C.Z., J.H. and E.L. performed the numerical calculation. All coauthors discussed the results. H.B.Z. and H.C.Z. wrote the paper with contributions from all authors.

## Competing interests
The authors declare no competing interests.
