## [Peer Review File · Nature Communications]

Reviewers' Comments:

Reviewer #1:

Remarks to the Author:

This is a timely, detailed, and scientifically sound study of the Voigt effect in a non-collinear antiferromagnet Mn₃Sn. Remarkably, the polarization rotation in the transient dynamic measurement is significantly larger than the MOKE induced rotation, or the non-magnetic contribution to the signal. The results are novel and highly relevant for the research of ultra-fast optical manipulation and detection antiferromagnets. Also, the study is very systematic, including the measured temperature dependence which provides a strong evidence that the signal originates from the Voigt effect of the non-collinear spins. The authors also provide a detailed modelling of their data which is consistent with experimental observations. I recommend the paper for publication.

Reviewer #2:

Remarks to the Author:

The authors present a time-resolved study of MO effect in a noncollinear antiferromagnet Mn₃Sn. By using an epitaxial (11-20) thin film sample with the Kagome bilayer aligned normal to the sample surface, they observed the ultrafast modulation of the polarization of the probe pulse and attribute it to the Voigt effect. The dependences of the signal on the sample angle, temperature, and magnetic field are explained by the Voigt effect. In contrast to conventional magnets, the quench dynamics in Mn₃Sn was found to be not sensitive to temperature close to T_N. The authors attributed it to the weakened DM interaction.

The large MO signal of Mn₃Sn is important for the field of spintronics and the presented data set of the Voigt effect look solid and new. But at this moment I have still some comments and questions.

(1) Figure 4 shows that the large ΔP signal of 0.3 mrad is dominated by the Voigt effect, and the MOKE signal contribution is as small as 0.02 mrad. Then the authors concluded that the Voigt signal is at least one order of magnitude larger than the MOKE and explained it as the Voigt signal is intrinsically larger than the MOKE.

If I understand it correctly, this comparison sounds odd because the magnetization direction in this manuscript is in-plane on the sample surface. Usually, the MOKE signal is determined by the magnetization component parallel to the light wavevector. In the in-plane configuration, the MOKE signal becomes small. I do not understand what this comparison means.

Mn₃Sn has attracted many recent attentions because of the large anomalous responses, and one of them is the large polar MOKE signal as large as 20 mdeg (~ 0.35 mrad) in bulk reported in Ref. 18. In addition, the MOKE signal is dependent on wavelength. Although the authors concluded that the Voigt signal is intrinsically much larger than the MOKE, I feel that it is only in the limited configuration and wavelength. The authors should explain this comparison more convincingly.

(2) The introduction in this manuscript begins with the interest for faster-speed information storage technology using AFM and the difficulty of detecting the AFM order. Then the authors investigated the Voigt effect of Mn₃Sn because it remains unexplored in noncollinear AFMs. But in the conclusion part, they only refer to the large Voigt signal and temperature-insensitive quench dynamics. Readers cannot understand how the finding in this manuscript is related to the interest in the introduction. The temperature-insensitive dynamics near T_N related to the DM interaction sounds interesting for fundamental physics but seems not sufficient for the significance required for publication in Nature Communications. The authors should explicitly show how their findings are important along with the context of the introduction.

(3) The explanation of the experiment is not easy to understand.

The authors describe the configurations of probe laser and magnetic field as "horizontal." It is not a good description because the horizontal plane is defined in the laboratory coordinate. In the schematic in Fig.1(b), however, the authors only define the xyz coordinate. Strictly speaking, readers do not determine whether the sample surface is set horizontally or vertically.

The lines 78-80 in the main text, "the magnetic moments projected on the surface plane in

samples with the (0001) plane normal to the sample surface can be viewed as collinear alignment, analogous to the collinear antiferromagnets" are difficult to read. The authors should explicitly describe how the magnetic moment is aligned in the experimental setup by using a figure. θ and ϕ in Eq. (1) should be also clearly indicated in the schematic to explain the Voigt effect for the clarity.

(4) θ -dependence of ΔR was shown only in Supplementary information and just described as "keeps nearly constant". The authors should plot the θ -dependence of the peak height of ΔR in Fig. 2(b) and directly compare with ΔP , as they did in Figs. 3(c) and 3(f).

Reviewer #3:

The manuscript from Zhao et al. addresses measurements of the ultrafast spin dynamics in the non collinear antiferromagnet Mn_3Sn . Systematic experiments as a function of temperature, polarization, magnetic field and sample orientation allow the authors to solidly assert that the detected rotation of polarization is an expression of the magneto-optical Voigt effect. The action of the laser pulses on the magnetic structure is equivalent to ultrafast heating, conform with the metallic nature of the sample, which induces a femtosecond reduction of the sublattices magnetisation and antiferromagnetic vector. A discussion of the symmetry of the Voigt effect is reported, also supported by phenomenological theory and simulations in the supplementary materials.

I believe the paragraph above to be a fair and complete synopsis of the current manuscript: all of this is textbook physics nowadays, since every one of these point has been discussed in papers, even in reviews, published 10 or more years ago, as I am going to elaborate in the following. The manuscript is far from being clear in the description of the experimental methods and, most importantly, does not motivate some very peculiar experimental choices. The use of the literature is not adequate, since many relevant works, which have been major results in the field, have been neglected. Overall, the proper audience for this work is the highly-specialised technical community working on time-resolved experiments in antiferromagnets. Thus, an improved and corrected version of this manuscript, could be suitable for publication in PRB or in New Journal of physics, but for sure neither for the readership of Nature Communication not of Scientific Reports.

Detailed report and technical points

- The authors properly motivate the research field of ultrafast magnetism and ultrafast spintronics. However, the present work is motivated by the statement: “However, the effectiveness of Voigt effect in detecting AFM spin dynamics in antiferromagnets with noncollinear spin alignment is yet to be demonstrated.” This is not true. The time-resolved Voigt effect on the femtosecond time-scale in a non collinear antiferromagnet was experimentally demonstrated and thoroughly analysed, with a similar approach to what the authors employ in the supplementary materials, in 2008 (PRB **78**, 104301, (2008)). Moreover, the entire discussion of the nature of the Voigt effect on the ultrafast timescale (temperature dependence, dependence on the sample’s orientation and magnetic field) has been already reported for a wide class of materials in a variety of papers NOT CITED by the authors (PRB **78**, 104301, (2008), PRB **89**, 060405(R) (2014), PRB **95**, 174407 (2017), PRB **84**, 064402 (2011), just to cite a few of them), including review papers (Review of Modern Physics **82**, 2731–2784 (2010), *Physica Scripta* **92**, 024002 (2017), Physics Uspekhi **58**, (10) 969 - 980 (2015)), the first of which has been cited more than 900 times. This point certifies both the lack of novelty and impact of the present paper, and the inappropriate use of the literature as well.
- The authors change the angle of the sample. Which configuration is shown in Fig. 1(b)? $\theta = 90$ deg, $\theta = 0$ deg or none of these two? The angle θ is not indicated Fig. 1(b). This point should be clarified.
- The author measure the dependence of the signal on the probe polarization by rotating the analyser. The optical dynamics, which the authors properly measure and compare with the spin dynamics, is measured by removing the analyser. This is highly unconventional, so: i) why do the authors not rotate the polarization of the probe beam? This measurement requires to change only one experimental parameter (probe polarization by rotating a waveplate) instead of two (angle of the analyser and angle of the sample). Moreover the canonical approach (i.e. rotating the probe polarization)

can be adapted also when the sample is in the cryostat. ii) why do the authors not employ the standard balanced detection? The standard scheme, with two lock-in amplifiers, allows to measure SIMULTANEOUSLY the rotation and the reflectivity, under the very same experimental conditions, which is clearly advantageous.

I do not claim that the choices of the authors are worse than the typical experimental approach, but since they are so unconventional and, at least apparently, a bit more limited, I think they should be discussed.

- Why is the transient reflectivity not normalised on the static value? Typically data are shown in units of $\Delta R/R$ and not just ΔR .
- Why is it impossible to derive an analytical formula for the Voigt effect in the case of an hexagonal lattice? Once the proper tensor components are known, as they are for all crystal groups, it should be possible to write a set of equations for each symmetry. I am surely missing something here, so the authors are welcome to explain me this point.
- Along which direction is the magnetic field applied?
- At the end of the paper, the authors describe the lack of critical slowing down in their data, as compared to “conventional FM and AFM oxides”. First, what is a “conventional AFM oxide”? Their sample seems magnetically very conventional as well, although it is not an oxide. Second, the authors fit the data and claim that they observe a “slightly longer” characteristic time for the fast relaxation as temperature is increased close to T_N . I would like to comment that they analyse data at a value of temperature 50 K away from T_N , which is very difficult to understand for me, since they measured at temperature approaching T_N (see Fig. 3 for instance). Why did they not use the 450 K for the discussion of the critical regime? Moreover, the claim of “slightly longer” time-scale is not corroborated by the data and the data analysis performed by the authors themselves: $\tau_{300\text{ K}} = 230\text{ fs} \pm 20$ and $\tau_{410\text{ K}} = 260 \pm 30\text{ fs}$ are consistent within the errorbars. So the authors report NO change in the time constant.

Response letter (NCOMMS-21-03322)

Dear Editor:

We sincerely thank the reviewers for the valuable suggestions and comments on our manuscript. These remarks are very helpful for us to improve the manuscript. In this response letter, we have addressed the concerns and questions raised by the reviewers. The manuscript has been revised accordingly and changes in the revised manuscript have been highlighted in yellow.

In the following, we have included the responses to the comments highlighted by Editor, and summary of changes in the revised manuscript, followed by the point-to-point responses to the comments of the reviewers.

Responses to the comments highlighted by Editor

You will notice in particular that reviewer three points to a large body of literature already covering the topic of your manuscript, which they claim limits the novelty of your work. In your response to reviewers, please clearly explain the advance of your manuscript over the works mentioned by the reviewer.

Reply: The main topics in the literature mentioned by reviewer three actually have essential differences with the current manuscript. These literatures report on the photoinduced coherent magnon motion in two-sublattice antiferromagnets. Reviewer #3 calls those antiferromagnets as “noncollinear antiferromagnets” just because the magnetic moments of two sublattice have a canting of $\sim 1^\circ$, but they essentially belong to the collinear antiferromagnets. In those works, the magnetic linear birefringence used to detect the spin precession motion is still described by collinear antiferromagnetic order L . This is an intrinsic difference with the three-sublattice antiferromagnet like Mn_3Sn , where the inverse triangular spin arrangement cannot be described by a collinear parameter L . The genuine noncollinear spin structure is exactly the reason that Mn_3Sn exhibits exotic properties, including the large anomalous Hall effect and magneto-optical Kerr effect unrelated to the net magnetic moments, which do not exist in those two sublattice antiferromagnets. Our motivation is to extend the application of the Voigt effect in detecting noncollinear spin order like inverse triangular spin order, and study the ultrafast spin process in the three-sublattice antiferromagnet. Detailed explanations have been included in the section of point-to-point responses.

Summary of changes

1. We have clarified the descriptions of the configurations of probe laser beam path and

magnetic field by using the xyz coordinate, in the Result section and Method section.

2. We have revised Fig. 1 to clarify how the magnetic moment and sample orientation are aligned in the experimental setup.
3. We have summarized the significance of our findings in the conclusion part, along with the context of the introduction.
4. We have revised Fig. 2(b) to directly compare the sample orientation dependence of ΔP and ΔR dynamics.
5. We have added new references [14-19] mentioned by Reviewer three and described the difference between these new references and current work in the introduction.
6. We have revised the title of Y-axis in transient reflectivity data figures (in Fig. 1, 2, 3, 5 and supplementary Fig. 4) to $\Delta R/R$.
7. We have clarified the description of the experimental method, including the Voigt signal measurements by rotating sample in oblique incidence configuration.
8. We have included the spin dynamics at 430 K for the comparison of the fast demagnetization.
9. We have revised the expressions throughout the paper to convey our perspectives more accurately, including the descriptions of “conventional AFM” and “slightly longer” time constant of spin dynamics.

Point-to-point responses:

Reviewer #1:

This is a timely, detailed, and scientifically sound study of the Voigt effect in a non-collinear antiferromagnet Mn₃Sn. Remarkably, the polarization rotation in the transient dynamic measurement is significantly larger than the MOKE induced rotation, or the non-magnetic contribution to the signal. The results are novel and highly relevant for the research of ultra-fast optical manipulation and detection antiferromagnets. Also, the study is very systematic, including the measured temperature dependence which provides a strong evidence that the signal originates from the Voigt effect of the non-collinear spins. The authors also provide a detailed modelling of their data which is consistent with experimental observations. I recommend the paper for publication.

Reply: We thank the reviewer #1 for his/her careful reading and very positive comments of our manuscript.

Reviewer #2:

The authors present a time-resolved study of MO effect in a noncollinear antiferromagnet Mn₃Sn. By using an epitaxial (11-20) thin film sample with the Kagome bilayer aligned normal to the sample surface, they observed the ultrafast modulation of the polarization of the probe pulse and attribute it to the Voigt effect. The dependences of the signal on the sample angle, temperature, and magnetic field are explained by the Voigt effect. In contrast to conventional magnets, the quench dynamics in Mn₃Sn was found to be not sensitive to temperature close to T_N. The authors attributed it to the weakened DM interaction.

The large MO signal of Mn₃Sn is important for the field of spintronics and the presented data set of the Voigt effect look solid and new. But at this moment I have still some comments and questions.

Reply: We thank the reviewer #2 for his/her valuable comments of our manuscript. According to those comments, we have carefully revised our manuscript.

Q1. Figure 4 shows that the large ΔP signal of 0.3 mrad is dominated by the Voigt effect, and the MOKE signal contribution is as small as 0.02 mrad. Then the authors concluded that the Voigt signal is at least one order of magnitude larger than the MOKE and explained it as the Voigt signal is intrinsically larger than the MOKE.

If I understand it correctly, this comparison sounds odd because the magnetization direction in this manuscript is in-plane on the sample surface. Usually, the MOKE signal is determined by the magnetization component parallel to the light wavevector. In the in-plane configuration, the MOKE signal becomes small. I do not understand what this comparison means.

Mn₃Sn has attracted many recent attentions because of the large anomalous responses, and one of them is the large polar MOKE signal as large as 20 mdeg (~0.35 mrad) in bulk reported in Ref. 18. In addition, the MOKE signal is dependent on wavelength. Although the authors concluded that the Voigt signal is intrinsically much larger than the MOKE, I feel that it is only in the limited configuration and wavelength. The authors should explain this comparison more convincingly.

Reply: We thank the reviewer for this constructive comment. First, we would like to clarify the experimental configuration of the magneto-optical effects. For Mn₃Sn with three spin-sublattices, the Mn magnetic moments form the noncollinear antiferromagnetic (AFM) ordering of the inverse triangular structure in the kagome plane, which is normal to the sample surface in our experiments. This AFM order is detected by the Voigt effect. Meanwhile, the

inverse triangular spin structure breaks the time-inversion symmetry and induces the ferroic ordering, which leads to the longitudinal MOKE signals in the absence of net magnetization. To detect in-plane AFM and ferroic order simultaneously, the probe light was applied under an incident angle of 40° . It is a typical experimental configuration of the longitudinal MOKE, which may change sign under the opposite in-plane magnetic fields exerted in our experiments. The intrinsic longitudinal MOKE signal is not small in this configuration and actually it is comparable to the polar MOKE signal, as can be seen in the reviewer mentioned Ref. 18, where the large longitudinal MOKE signal (~ 0.25 mrad), at the incident angle of 45° , is reported in a bulk Mn_3Sn with its kagome plane normal to the sample surface. Therefore, the direct comparison of the transient Voigt and MOKE signals in this configuration can truly reflect their intrinsic magnitude modulated by the laser excitation. In our work, the observed modulated MOKE signal is only about 0.02 mrad, at least one order of magnitude smaller than the modulated Voigt signal, so we just claim that the modulated Voigt angle is significantly larger than the modulated MO Kerr angle arising from the ferroic ordering of cluster magnetic octupole. To be more accurately, we have added the description of the wavelength of probe light (800 nm) for this comparison in the revised manuscript.

Q2. The introduction in this manuscript begins with the interest for faster-speed information storage technology using AFM and the difficulty of detecting the AFM order. Then the authors investigated the Voigt effect of Mn_3Sn because it remains unexplored in noncollinear AFMs. But in the conclusion part, they only refer to the large Voigt signal and temperature-insensitive quench dynamics. Readers cannot understand how the finding in this manuscript is related to the interest in the introduction. The temperature-insensitive dynamics near T_N related to the DM interaction sounds interesting for fundamental physics but seems not sufficient for the significance required for publication in Nature Communications. The authors should explicitly show how their findings are important along with the context of the introduction.

Reply: We thank the reviewer for this important suggestion. The importance of the findings has been summarized in the conclusion part in the revised manuscript. Firstly, the current work experimentally and theoretically studies the Voigt effect in Mn_3Sn with noncollinear AFM order, which opens a new and easily accessible avenue to study ultrafast spin dynamics in noncollinear antiferromagnets. The method of detecting noncollinear AFM order with Voigt effect can be extended to a large amount of AFM materials. Secondly, the temperature-insensitive spin dynamics is significant for the fast recording in storage devices like heat-assisted magnetic recording (HAMR). Generally, it is required in those devices to

heat the magnetic media close to its Curie temperature (T_c) during recording in order to make the coercivity smaller than the applied magnetic field. Even for laser helicity-dependent magnetic switching without applied field, the deterministic switching occurs only when the media is heated close to T_c . When the temperature is close to T_c , the spin motion in conventional ferromagnets becomes significantly slower (see in Refs. 34-36), which increases the magnetization reversal time. In the current findings, the spin motion in noncollinear antiferromagnet keeps fast near T_N , which is beneficial for the ultrafast manipulation of magnetic order and therefore high speed recording in storage devices. We have included in the conclusion of the revised manuscript more discussions of the significance of our work along with the context of the introduction.

Q3. The explanation of the experiment is not easy to understand.

The authors describe the configurations of probe laser and magnetic field as “horizontal.” It is not a good description because the horizontal plane is defined in the laboratory coordinate. In the schematic in Fig.1(b), however, the authors only define the xyz coordinate. Strictly speaking, readers do not determine whether the sample surface is set horizontally or vertically. The lines 78-80 in the main text, “the magnetic moments projected on the surface plane in samples with the (0001) plane normal to the sample surface can be viewed as collinear alignment, analogous to the collinear antiferromagnets” are difficult to read.

The authors should explicitly describe how the magnetic moment is aligned in the experimental setup by using a figure. θ and ϕ in Eq. (1) should be also clearly indicated in the schematic to explain the Voigt effect for the clarity.

Reply: We thank the reviewer for pointing out the unclear descriptions of the experiment in our manuscript.

- (i) In the experimental setup, the incident and reflected laser beam and magnetic field are in the “horizontal” plane, which is parallel to the xz plane. Meanwhile the sample surface is aligned parallel to the xy plane, which represents the “vertical” plane. To be more clearly, we have revised the description of the configurations of probe laser beam path and magnetic field by using the xyz coordinate. The complete description of the experimental configuration is as follows: the sample surface normal is defined as the z axis; the incident plane of probe laser is parallel to the horizontal xz plane; and the magnetic field is applied along the x axis. We have changed the texts in the lines 78-80 as follows: “The inverse triangular AFM spins in Mn_3Sn align within the (0001) plane. For Mn_3Sn films with the (0001) plane normal to the film surface, the in-plane projection of the spins is arranged

only in the direction perpendicular to the [0001] axis along the surface. In this case, the in-plane magnetic moments can be viewed as collinear alignment, analogous to the collinear antiferromagnets.”

- (ii) We have revised Fig. 1 to clarify how the magnetic moment is aligned in the experimental setup. In Fig. 1(a), the magnetic moments are illustrated with respect to the [0001], $[\bar{1}100]$, and $[11\bar{2}0]$ crystal axes. In Fig. 1(b), the sample and experimental configuration is described along with the xyz coordinate system. The $[\bar{1}100]$ and [0001] axes are along the sample surface and the $[11\bar{2}0]$ axis is parallel to the sample surface normal. The sample is placed on a motorized rotation stage with its surface parallel to the vertical xy plane and the $[11\bar{2}0]$ axis (surface normal) along the z axis. When the sample is rotated around the z axis, we introduce θ to denote the angle between the Mn_3Sn $[\bar{1}100]$ direction and the horizontal x axis, as indicated in the inset of Fig. 1(c). θ is used in Eq. (1) to also describe the orientation of the projected AFM spins on the surface plane. In Eq. (1), φ represents the angle between the orientation of probe light polarization and the x axis. In the experiment, the probe laser is s -polarized light, so φ is 90° . In the revised manuscript, φ has been indicated in Fig. 1(b).

Q4. θ -dependence of ΔR was shown only in Supplementary information and just described as “keeps nearly constant”. The authors should plot the θ -dependence of the peak height of ΔR in Fig. 2(b) and directly compare with ΔP , as they did in Figs. 3(c) and 3(f).

Reply: We thank the reviewer for this suggestion. The peak ΔR values have been extracted from the dynamic curves shown in Fig. S4 in Supplementary information. To directly compare ΔR and ΔP , the normalized peak $\Delta R/R$ and peak ΔP have been included in Fig. 2(b) in the revised manuscript. The sentence in lines 188-189 has been revised to “For comparison, we also have measured the ΔR dynamics for different θ (shown in Supplementary Fig. 4). The overall ΔR dynamics and peak amplitudes of ΔR indicate no θ dependence, as shown in Fig. 2(b).”

Reviewer #3:

The manuscript from Zhao et al. addresses measurements of the ultrafast spin dynamics in the non collinear antiferromagnet Mn_3Sn . Systematic experiments as a function of temperature, polarization, magnetic field and sample orientation allow the authors to solidly assert that the

detected rotation of polarization is an expression of the magneto-optical Voigt effect. The action of the laser pulses on the magnetic structure is equivalent to ultrafast heating, conform with the metallic nature of the sample, which induces a femtosecond reduction of the sublattices magnetization and antiferromagnetic vector. A discussion of the symmetry of the Voigt effect is reported, also supported by phenomenological theory and simulations in the supplementary materials.

I believe the paragraph above to be a fair and complete synopsis of the current manuscript: all of this is textbook physics nowadays, since every one of these point has been discussed in papers, even in reviews, published 10 or more years ago, as I am going to elaborate in the following. The manuscript is far from being clear in the description of the experimental methods and, most importantly, does not motivate some very peculiar experimental choices. The use of the literature is not adequate, since many relevant works, which have been major results in the field, have been neglected. Overall, the proper audience for this work is the highly-specialised technical community working on time-resolved experiments in antiferromagnets. Thus, an improved and corrected version of this manuscript, could be suitable for publication in PRB or in New Journal of physics, but for sure neither for the readership of Nature Communication nor of Scientific Reports.

Reply: We thank the reviewer #3 for his/her careful review and critical comments. The reviewer comments that the points we presented in the submitted manuscript have been discussed in published papers, even in reviews. We would like to further address that we demonstrate for the first time to apply the Voigt effect for detection of the ultrafast spin dynamics in a genuine noncollinear AFM material Mn_3Sn , which is essentially different with nominal noncollinear AFM materials of two sublattice magnetization with slight canting in all published papers. The DMI interaction plays an important role in forming the inverse triangle spin structure in Mn_3Sn , and it breaks the time inversion symmetry and induces MOKE signal unrelated to the net magnetization, which is very unique compared to the AFM materials studied for the ultrafast spin dynamics in the literature. The large MOKE signal in Mn_3Sn has intrigued significant interests recently, in spite of the negligible net magnetization. As an important step forward, we found the modulated Voigt signal is tremendously larger than the modulated MOKE signals in Mn_3Sn . Moreover, we found that the AFM spin modulation shows no slowing down with rising temperature close the magnetic transition temperature. These new findings are significant because it indicates that the modulation of noncollinear AFM order maintained by strong DMI interaction can keep fast near T_N , in contrast to the modulation of FM and collinear AFM order governed by the exchange coupling. This is beneficial for the

ultrafast manipulation of magnetic order and therefore high speed recording in storage devices. In the following, we have given the point-to-point responses to the reviewer's questions and comments, and the manuscript has been revised accordingly. We hope that our responses have addressed all the concerns of the reviewer.

Q1. The authors properly motivate the research field of ultrafast magnetism and ultrafast spintronics. However, the present work is motivated by the statement: "However, the effectiveness of Voigt effect in detecting AFM spin dynamics in antiferromagnets with noncollinear spin alignment is yet to be demonstrated." This is not true. The time-resolved Voigt effect on the femtosecond time-scale in a noncollinear antiferromagnet was experimentally demonstrated and thoroughly analysed, with a similar approach to what the authors employ in the supplementary materials, in 2008 (PRB 78, 104301, (2008)). Moreover, the entire discussion of the nature of the Voigt effect on the ultrafast timescale (temperature dependence, dependence on the sample's orientation and magnetic field) has been already reported for a wide class of materials in a variety of papers NOT CITED by the authors (PRB 78, 104301, (2008), PRB 89, 060405(R) (2014), PRB 95, 174407 (2017), PRB 84, 064402 (2011), just to cite a few of them), including review papers (Review of Modern Physics 82, 2731–2784 (2010), Physica Scripta 92, 024002 (2017), Physics Uspekhi 58, (10) 969 - 980 (2015)), the first of which has been cited more than 900 times. This point certifies both the lack of novelty and impact of the present paper, and the inappropriate use of the literature as well.

Reply: We thank the reviewer for this critical comment. We would like to further clarify the motivation of the current work and the differences from the literature mentioned above. The paper (PRB 78, 104301, (2008)) presents the optically excited spin precession in weak ferromagnet FeBO₃ using magnetic linear birefringence (MLB). FeBO₃ has a bipartite lattice with antiferromagnetically coupled Fe moments M_1 and M_2 . In this paper, magnetic structure is described by the ferromagnetic vector $M=M_1+M_2$ and the collinear antiferromagnetic vector $L=M_1-M_2$. The main purpose of this paper is to experimentally and theoretically study the excitation of magnons and phonons by optical pulses. The MLB used in this paper is related to the vector L , equivalent to the collinear AFM order. From this point, spin alignment in two-sublattice antiferromagnets should be viewed as collinear configuration. Strictly speaking, compensated antiferromagnets have noncollinear spin alignment only when the antiferromagnets have odd sublattices. In this case, the magnetic moments have noncollinear local easy axis. For instance, Mn₃Sn with three spin-sublattices forms inverse triangular spin

structure. There is yet no demonstration for the effectiveness of MLB or Voigt effect in detecting AFM spin dynamics in noncollinear antiferromagnets with inverse triangular spin alignment, since traditional description of MLB or Voigt effect is based on the collinear AFM vector L .

In addition, the paper (PRB 89, 060405(R) (2014)) reports on the photo energy dependence of spin dynamics excited by the light-magnon interaction without heating of electrons or phonons; the paper (PRB 95, 174407 (2017)) presents the detection of the magnon oscillations in NiO excited via IFE and ICME, and derived the formulas of FE, CME, IFE and ICME using M and L related dielectric tensor; the paper (PRB 84, 064402 (2011)) mainly deals with the wavelength dependence of spin precession in DyFeO₃ induced via ISRS. The main concern of these studies is the photoinduced coherent magnons motion, which is described by M and L . The current work is motivated by the effectiveness of Voigt effect in detecting antiferromagnetic order with inverse triangular spin alignment, which cannot be described by M and L . The review papers (Review of Modern Physics 82, 2731–2784 (2010), Physica Scripta 92, 024002 (2017), Physics Uspekhi 58, (10) 969 - 980 (2015)) cover the topic of antiferromagnetic spintronics, but we believe that there is a lack of the discussion about the Voigt effect in noncollinear antiferromagnet with odd spin-sublattices.

We thank the reviewer for pointing out those studies of AFM spin dynamics to us. To be more adequate in introducing the progress of ultrafast AFM spin dynamics research, the papers mentioned by the reviewer have been cited in the introduction part. The difference between those papers and current work has also been described in the introduction.

Q2. The authors change the angle of the sample. Which configuration is shown in Fig. 1(b)? $\theta = 90$ deg, $\theta = 0$ deg or none of these two? The angle θ is not indicated Fig. 1(b). This point should be clarified.

Reply: We thank the reviewer for pointing out this unclear point in our manuscript. The experimental schematic shown in Fig. 1(b) corresponds to the sample configuration with $\theta=0^\circ$. In the revised manuscript, this configuration has been clarified in the caption of Fig. 1(b). For measurements of rotating sample, θ is indicated in the inset of Fig. 1(c).

Q3. The author measure the dependence of the signal on the probe polarization by rotating the analyser. The optical dynamics, which the authors properly measure and compare with the spin dynamics, is measured by removing the analyser. This is highly unconventional, so: i) why do the authors not rotate the polarization of the probe beam? This measurement requires to change

only one experimental parameter (probe polarization by rotating a waveplate) instead of two (angle of the analyser and angle of the sample). Moreover the canonical approach (i.e. rotating the probe polarization) can be adapted also when the sample is in the cryostat. ii) why do the authors not employ the standard balanced detection? The standard scheme, with two lock-in amplifiers, allows to measure SIMULTANEOUSLY the rotation and the reflectivity, under the very same experimental conditions, which is clearly advantageous.

I do not claim that the choices of the authors are worse than the typical experimental approach, but since they are so unconventional and, at least apparently, a bit more limited, I think they should be discussed.

Reply: We feel sorry for the unclear descriptions of the experimental methods. The reason for rotating the sample instead of the polarization of the probe is as follows. For Mn_3Sn with the inverse triangular spin structure, the magnetic moments show AFM ordering in the Kagome plane, meanwhile it exhibits longitudinal MOKE due to the ferroic ordering of cluster magnetic octupole. To detect the AFM order via Voigt effect and the ferroic ordering via longitudinal MOKE simultaneously, we steer the probe light at a finite incident angle of $\sim 40^\circ$. In this configuration, we may apply in-plane field to switch the ferroic ordering and change the sign of MOKE signals. However, in the case of oblique incidence, it is not appropriate to rotate the polarization of the probe laser when measuring the polarization rotation of the MO effect, since the different reflection of light components with s - and p - polarization can induce the polarization rotation, even in non-magnetic samples, according to the Fresnel formula. Therefore, the probe light was kept in the s -polarized state in the experiment. Actually, the polarization rotation measurements by rotating sample in our oblique incidence configuration can also help to distinguish the contributions from the AFM order or from the net magnetization. The net magnetization would lead to linear MOKE signals with the sinusoidal form of 360° periodicity when rotating the sample, different from the AFM order induced signals. On the contrary, the net magnetization would result in Voigt signals with similar symmetry to that originated from the AFM order if the polarization rotation is measured by rotating the polarization of the probe light in the configuration of perpendicular incidence.

As the reviewer mentioned, balanced detection can be used in the detection of the polarization rotation and it shows much better signal to noise ratio for pulsed laser system with high repetition rate. However, according to our test, for the ultrafast laser with 1 kHz repetition rate used in our experiment, balanced detection is more effective only for the extremely weak probe light. For the moderate probe pulse intensity ($>0.1 \mu\text{J}$), the balanced detection does not

show clear advantages of signal to noise ratio over the method of crossed polarizers. So we just used crossed polarizers for the measurements of polarization rotation. Using this method can help us better monitor and control the extinction. In the measurements of sample orientation dependence, we actually do not need to rotate the analyzer. In Fig. 1(c), we show the results of different analyzer angles opposite to the extinction just for providing additional evidence that the measured transient signals correspond to the polarization rotation.

Q4. Why is the transient reflectivity not normalised on the static value? Typically data are shown in units of $\Delta R/R$ and not just ΔR .

Reply: We thank the reviewer for pointing out this issue. The transient reflectivity data shown in the previous manuscript had actually been normalized on the static value. The title of Y-axis in transient reflectivity data figures should be $\Delta R/R$. It was a mistake and we have corrected it.

Q5. Why is it impossible to derive an analytical formula for the Voigt effect in the case of an hexagonal lattice? Once the proper tensor components are known, as they are for all crystal groups, it should be possible to write a set of equations for each symmetry. I am surely missing something here, so the authors are welcome to explain me this point.

Reply: The analytical formula for the Voigt effect or MLB can be derived by solving Maxwell equations $\nabla \times (\nabla \times E) = -c^{-2} \varepsilon \frac{\partial^2 E}{\partial t^2}$. First, we consider a simple case that the wave ($E = E_0 \exp(i\omega t - ikr)$) is normally incident on the sample with cubic lattice. The wave vector is $k = \frac{\omega}{c} n_z \mathbf{z}$. The symmetrical part of ε defines the value of the Voigt effect or MLB. Then the wave equation is

$$\begin{pmatrix} \frac{\partial^2}{\partial z^2} & 0 & 0 \\ 0 & \frac{\partial^2}{\partial z^2} & 0 \\ 0 & 0 & 0 \end{pmatrix} \begin{pmatrix} E_{0x} \\ E_{0y} \\ E_{0z} \end{pmatrix} = \omega^2 c^{-2} \begin{pmatrix} \varepsilon_{xx} & \varepsilon_{xy} & 0 \\ \varepsilon_{xy} & \varepsilon_{xx} & 0 \\ 0 & 0 & \varepsilon_{xx} \end{pmatrix} \frac{\partial^2}{\partial t^2} \begin{pmatrix} E_{0x} \\ E_{0y} \\ E_{0z} \end{pmatrix} \quad (\text{R1})$$

From Eq. (R1), it is easy to get two eigenvalues $n_z^\pm = \sqrt{\varepsilon_{xx} \pm \varepsilon_{xy}}$. In the transmission case, over a propagation distance d , the linear birefringence induces a phase difference of $\Phi = \frac{\omega d}{c} (n_z^+ - n_z^-) \approx \frac{\omega d \varepsilon_{xy}}{c \sqrt{\varepsilon_{xx}}}$.

Now, we consider a more complicated case that the wave is incident on the sample with hexagonal lattice at a finite incidence angle φ . The plane of incidence is parallel to xz plane. So the wave vector is $k = \frac{\omega}{c} (n_x \mathbf{x} + n_z \mathbf{z})$. For our Mn_3Sn sample, the hexagonal c axis lies in the surface plane. Then the wave equation becomes

$$\begin{pmatrix} \frac{\partial^2}{\partial z^2} & 0 & -\frac{\partial^2}{\partial x \partial z} \\ 0 & \left(\frac{\partial^2}{\partial z^2} + \frac{\partial^2}{\partial x^2}\right) & 0 \\ -\frac{\partial^2}{\partial z \partial x} & 0 & \frac{\partial^2}{\partial x^2} \end{pmatrix} \begin{pmatrix} E_{0x} \\ E_{0y} \\ E_{0z} \end{pmatrix} = \omega^2 c^{-2} \begin{pmatrix} \varepsilon_{xx} & \varepsilon_{xy} & 0 \\ \varepsilon_{xy} & \varepsilon_{yy} & 0 \\ 0 & 0 & \varepsilon_{xx} \end{pmatrix} \frac{\partial^2}{\partial t^2} \begin{pmatrix} E_{0x} \\ E_{0y} \\ E_{0z} \end{pmatrix} \quad (\text{R2})$$

The proper value equation for the refractive index follows from the determinant condition of Eq. (R2), which is expressed as

$$\varepsilon_{xx} n_z^4 + (2\varepsilon_{xx} n_x^2 - \varepsilon_{xx} \varepsilon_{yy} - \varepsilon_{xx}^2) n_z^2 + [(\varepsilon_{xx} n_x^2 - \varepsilon_{xx}^2)(n_x^2 - \varepsilon_{yy}) + \varepsilon_{xy}^2 (n_x^2 - \varepsilon_{xx})] = 0 \quad (\text{R3})$$

In the reflection cases, one can get the polarization rotation of incident light by the following steps: first get two eigenvalues of n_z and corresponding amplitude components E_{0x} , E_{0y} , E_{0z} , and then get the reflection coefficients based on the continuity conditions. However, the solutions of Eq. (R3) cannot be simplified to a concise analytical expression. Therefore, we gave a series of equations to express the reflection coefficients in hexagonal AFM sample in the Supplementary Information, and then we used numerical simulations to analyze the Voigt effect.

In conclusion, the unequal diagonal elements ($\varepsilon_{xx} \neq \varepsilon_{yy}$) and finite incidence angle restrict a clear analytical formula for the Voigt effect.

Q6. Along which direction is the magnetic field applied?

Reply: The magnetic field was applied along the x axis. We have clarified it in the revised manuscript.

Q7. At the end of the paper, the authors describe the lack of critical slowing down in their data, as compared to “conventional FM and AFM oxides”. First, what is a “conventional AFM oxide”? Their sample seems magnetically very conventional as well, although it is not an oxide. Second, the authors fit the data and claim that they observe a “slightly longer” characteristic time for the fast relaxation as temperature is increased close to TN. I would like to comment that they analyse data at a value of temperature 50 K away from TN, which is very difficult to understand for me, since they measured at temperature approaching TN (see Fig. 3 for instance). Why did they not use the 450 K for the discussion of the critical regime? Moreover, the claim of “slightly longer” timescale is not corroborated by the data and the data analysis performed by the authors themselves: $\tau_{300 \text{ K}} = 230 \text{ fs} \pm 20$ and $\tau_{410 \text{ K}} = 260 \pm 30 \text{ fs}$ are consistent within the errorbars. So the authors report NO change in the time constant.

Reply: We thank the reviewer for this constructive comment. First, compared with

“conventional AFM”, Mn_3Sn has some exotic properties like large AHE and large MOKE, although it has negligible net magnetic moments. Traditionally, collinear AFM materials with zero net magnetization don’t exhibit MOKE. Only because of the weak ferromagnetism resulted from the canting of two collinear spin-sublattice magnetization, they may exhibit MOKE. For Mn_3Sn , the large MOKE is originated from the ferroic ordering of cluster magnetic octupole formed by the inverse triangular AFM state, rather than from the small net FM moment as a result of canting of spin-sublattices. Therefore, Mn_3Sn with three-sublattice AFM structure seems unconventional compared with two-sublattice AFM materials. To be more clearly, we have revised “conventional AFM oxide” to “conventional two-sublattice AFM oxide”.

Secondly, the T_N of Mn_3Sn is around 420-430 K, so 410 K is already very close to T_N . The overall polarization signal at 410 K is much smaller than those below 400 K. Therefore, in the previous manuscript, the AFM spin dynamics at 300 K and 410 K were directly compared to demonstrate the change of time constants when T is increased close to T_N . As the reviewer suggested, a higher temperature near T_N is more suitable for the analysis. So we also have included the dynamics at 430 K for the comparison of the fast demagnetization. The demagnetization time constant at 430 is $\tau_{430\text{ K}} \approx 270 \pm 30$ fs, which is also comparable to $\tau_{300\text{ K}} \approx 230 \text{ fs} \pm 20$ at $T=300$ K. As the reviewer mentioned, the “slightly longer” timescale and the error bars are on the same order of magnitude. In the revised manuscript, we have used “negligible change” instead of “slightly longer” to avoid any confusion for the readers.

Yours sincerely,

Haibin Zhao, Ph. D., Professor
Dep. of Opt. Sci. & Engn.
Fudan Univ., Shanghai, China

Reviewers' Comments:

Reviewer #2:

Remarks to the Author:

The authors responded to my concerns carefully. They improved the explanation of experimental configuration and comparison with previous literature.

They also added how their findings are related to the potential interest for faster speed information storage technology. I strongly suggest that they should explicitly mention it in the abstract. Then, I recommend publication in this form.

Reviewer #3:

The revised version of the manuscript and the rebuttal have addressed the technical criticism in a clear and satisfactory way. I am personally glad to notice that the authors corrected their statements concerning the discussed variations of the time constant, previously found to be “slightly longer” and presently defined as “negligible change”.

Having said that, the main issue of this work is the novelty, which is incremental in view of the existing literature. In the rebuttal to my criticism, the authors convinced me even more about the incremental nature of their manuscript. In fact they write:

- *“The genuine noncollinear spin structure is exactly the reason that Mn₃Sn exhibits exotic properties, including the large anomalous Hall effect and magneto-optical Kerr effect unrelated to the net magnetic moments, which do not exist in those two sublattice antiferromagnets”.*

This is simply not true, since one of the samples already massively studied (FeBO₃) has a huge Faraday rotation (which is equivalent to the magneto-optical Kerr effect). This point is demonstrated experimentally in PRL **89**, 287401 (2002), as time-resolved Faraday rotations bigger than 8 mrad were measured. The Voigt effect reported by the authors amounts to 0.3 mrad, the MOKE signal they report is 0.02 mrad, while the literature reports longitudinal MOKE signal in Mn₃Sn as large as 0.25 mrad. The Faraday rotation detected in FeBO₃ is at least one order of magnitude higher than any magneto-optical effect in Mn₃Sn. The paper I cite here is 20 years old, cited more than 60 times but not cited by the authors. Taking this work into account, it is clear that Mn₃Sn does not possess all the exotic properties discussed by the authors and that it is a rather conventional system, which small magneto-optical activity, at least 10 times smaller than what is already reported in the literature. The microscopic origin of the effect (either magnetization or ferric order) is relevant in a discussion of the details which is of interest for specialists: the physically measured effect is the MOKE or Faraday rotation, which is 10 times bigger in already studied materials than in Mn₃Sn.

- *“These literatures report on the photoduced coherent magnon motion in two-sublattice antiferromagnets. Reviewer #3 calls those antiferromagnets as “noncollinear antiferramagnets” just because the magnetic moments of two sublattice have a canting of $\sim 1^\circ$. but they essentially belong to the collinear antiferromagnets.”*

This is once again not true: collinear antiferromagnets have zero Faraday rotation, while FeBO₃ shows a huge Faraday rotation, as discussed above. So weak ferromagnets are NOT collinear antiferromagnets.

The novelty of the present work cannot be the theory, since the authors have already presented it in a recently published work (AIP Advances **11**, 055003 (2021)). Furthermore the authors have already demonstrated that the Voigt effect can be employed to measure spin dynamics in antiferromagnets PRB **98**, 134409 (2018). The arguments provided in my previous report and here above show that the same conclusions (i.e. ultrafast spin dynamics detected by Voigt effect and even Faraday effect) were demonstrated in the literature 10-20 years ago also for non collinear systems, which are non collinear due to DMI, exactly like Mn₃Sn (as mentioned by the authors in the rebuttal).

I also would like to mention another very relevant paper: Nature Photonics **9**, 25–29 (2014). In this work it is shown that both the Faraday effect and the Voigt effect can trace the ultrafast spin

dynamics in YMnO₃, which is a noncollinear antiferromagnet with basically the same configuration as Mn₃Sn, i.e. three non collinear sublattices, in an hexagonal crystal structure with zero net magnetisation, as illustrated below (Fig. 1(b) and (c) from Nature Photonics **9**, 25–29 (2014)):

Given this discussion, the authors cannot claim that *“The effectiveness of Voigt effect in detecting AFM spin dynamics in antiferromagnets with noncollinear spin alignment is yet to be demonstrated.”*, as they still do in the introductory part of the manuscript. This effect has already been demonstrated in a wide variety of materials, including samples almost identical if not identical to Mn₃Sn (Nature Photonics **9**, 25–29 (2014)). Especially in view of Nature Photonics **9**, 25–29 (2014), the only novel information of the manuscript is that also in Mn₃Sn this spin dynamics can be detected by means of the Voigt effect. The difference in the dynamics, i.e. coherent magnons VS demagnetisation has nothing to do with the magnetic structure: it is a consequence of optically pumping either a metal (free electrons induced demagnetisation) or an insulator with photon energy in the bandgap (light-scattering driven magnons). This point is reported in the review papers now included in the revised version of the paper.

Thus the novelty reported by the authors is incremental and marginal, one more material can be measured as a lot of materials already investigated. This is hardly suitable for the broader audience of Nature Communications. I stand by my original suggestion and evaluation of this work, finding it proper for Phys. Rev. B or New Journal of Physics.

Response letter (NCOMMS-21-03322A)

Dear Editor:

We thank the reviewers again for giving their valuable suggestions and comments on our manuscript. Although we cannot agree with Referee #3 on his/her novelty concern of our work, the remarks of the reviewers are very helpful for us to further improve the discussion and clarification of our manuscript. In this response letter, we have addressed the concerns and questions raised by the reviewers. The manuscript has been revised accordingly and changes in the revised manuscript have been highlighted in yellow. We feel that the revised manuscript is now suitable for publication in Nature Communications.

In the following, we have included responses to the comments highlighted by Editor, the summary of changes in the revised manuscript, followed by the point-to-point responses to the comments of the reviewers.

Responses to the comments highlighted by Editor

The principle issue remains the proper presentation of prior work. You will see reviewer three draws attention to two papers, both of which provide results for antiferromagnets, one with an equivalently canted arrangement to Mn_3Sn . Please make sure, in your revisions that provide proper presentation of these earlier works, and establish a clear motivation for your work on Mn_3Sn .

Reply: Thank you for overseeing the review process and providing your evaluation of our work. We agree with you that our manuscript lacked proper presentation of prior works. As you will see in our revised manuscript and the response letter, we have carefully revised the presentation of motivation in the introduction part by including the references mentioned by review 3, and the differences of our work to these prior publications have been also clearly stated in the following response letter.

First, we remark that the comparison of large Faraday rotation obtained in $FeBO_3$ (300 μm thick) and Voigt/MOKE signals in Mn_3Sn (40 nm thick) is not straightforward. The Faraday effect measured in the transmission geometry is proportional to the sample thickness and our Voigt effect and MOKE measured in the reflection geometry are just surface sensitive. If roughly converting our Voigt/MOKE signals to the polarization rotations of the transmitted light in the Mn_3Sn sample with thickness of 300 μm by multiplying the thickness ratio (300/0.04), the signal in Mn_3Sn would be much larger than that in $FeBO_3$. The corresponding descriptions and reference [Ref. 28] have been included in the revised manuscript.

Second, we are also very confident our results are not incremental compared to

that on YMnO₃. The Voigt and Faraday signals in YMnO₃ are caused by the net magnetic moment related to the precessional spins which are NOT the intrinsic Voigt signals induced by the inherent antiferromagnetic (AFM) order we have focused on in Mn₃Sn and they cannot be used to investigate the ultrafast dynamics (quench) of the AFM order. In the revised manuscript, we have added the two new references [Refs. 20, 21] which report on the ultrafast spin dynamics in YMnO₃ with triangular spin structure, and included some descriptions to clearly distinguish its difference from our work. Accordingly, we have further clarified one of our motivations in the introduction: “For this class of noncollinear antiferromagnets with compensated spin alignment, the effectiveness of Voigt effect produced by the inherent noncollinear AFM order in detecting its ultrafast Néel order dynamics is yet to be demonstrated.” Moreover, the direct comparison of ultrafast Voigt and MOKE effects in Mn₃Sn and the understanding of ultrafast dynamics of the noncollinear Néel order are also important issues to be addressed since Mn₃Sn represents an important class of AFM materials.

We have also summarized the difference between our work and the literature in the following table to clarify the motivation and novelty of our present work.

FeBO₃: Slightly canted AFM spins, net magnetization induced transient Faraday rotation of ~8 mrad proportional to the crystal thickness of 300 μm, PRL 89, 287401 (2002). Ref. 28	Mn₃Sn:  ■ Compensated inverse triangular AFM spins; ■ Large static MOKE of ~0.25 mrad, surface sensitive, Nat. Photonics 12, 73 (2018); large Anomalous Hall effect, Nature 527, 212 (2015); Refs. 22&23 ■ Transient Voigt signal of ~0.3 mrad and transient MOKE signal of ~0.02 mrad in 40-nm Mn₃Sn, surface sensitive, this work; ■ Voigt effect induced by inherent noncollinear Néel order to monitor the ultrafast dynamics (quench) of Néel order, this work.
YMnO₃: Transient Voigt effect induced by symmetry broken in presence of net magnetization related to precessional spins excited by laser pulses, Nat. Photonics 9, 25 (2015). Ref. 20	
CuMnAs & CoO: Transient Voigt effect induced by collinear Néel order, Nat. Photonics 11, 91 (2017), PRB 98, 134409 (2018). Refs. 12&13	

Summary of changes

1. We have included the significance of our findings in the abstract.
2. We have added a new reference [20] mentioned by Reviewer #3 and another new

reference [21] of related work dealing with spin dynamics in a noncollinear antiferromagnet, and clarified the difference between these new references and current work in the introduction.

3. We have revised the expressions of the motivation of the current work in the introduction to convey our perspectives more clearly and accurately, including the description of the detection of “dynamics of the noncollinear AFM order”.

4. We have included a new reference [28] and corresponding descriptions on the size of Voigt/MOKE signals in Mn_3Sn and Faraday rotations in FeBO_3 .

Point-to-point responses:

Reviewer #2:

The authors responded to my concerns carefully. They improved the explanation of experimental configuration and comparison with previous literature.

They also added how their findings are related to the potential interest for faster speed information storage technology. I strongly suggest that they should explicitly mention it in the abstract.

Then, I recommend publication in this form.

Reply: We thank the reviewer for his/her very positive comments of our manuscript. According to the above suggestion, we have explicitly mentioned the relation of our finding to the potential interest for faster speed information storage technology in the abstract: “The temperature-insensitive ultrafast spin manipulation can pave the way for high-speed spintronic devices either working at a wide range of temperature or demanding spin switching near T_N .”

Reviewer #3:

The revised version of the manuscript and the rebuttal have addressed the technical criticism in a clear and satisfactory way. I am personally glad to notice that the authors corrected their statements concerning the discussed variations of the time constant, previously found to be “slightly longer” and presently defined as “negligible change”.

Reply: We thank the reviewer for the positive comments on our previous responses to the technical criticism. In the following, we further respond to the comments on the novelty of our work. According to those comments, we have carefully revised our descriptions of the motivation of our work.

Having said that, the main issue of this work is the novelty, which is incremental in view of the existing literature. In the rebuttal to my criticism, the authors convinced me even more about the incremental nature of their manuscript. In fact they write:

- *“The genuine noncollinear spin structure is exactly the reason that Mn₃Sn exhibits exotic properties, including the large anomalous Hall effect and magneto-optical Kerr effect unrelated to the net magnetic moments, which do not exist in those two sublattice antiferromagnets”. This is simply not true, since one of the samples already massively studied (FeBO₃) has a huge Faraday rotation (which is equivalent to the magneto-optical Kerr effect). This point is demonstrated experimentally in PRL 89, 287401 (2002), as time-resolved Faraday rotations bigger than 8 mrad were measured. The Voigt effect reported by the authors amounts to 0.3 mrad, the MOKE signal they report is 0.02 mrad, while the literature reports longitudinal MOKE signal in Mn₃Sn as large as 0.25 mrad. The Faraday rotation detected in FeBO₃ is at least one order of magnitude higher than any magneto-optical effect in Mn₃Sn. The paper I cite here is 20 years old, cited more than 60 times but not cited by the authors. Taking this work into account, it is clear that Mn₃Sn does not possess all the exotic properties discussed by the authors and that it is a rather conventional system, which small magneto-optical activity, at least 10 times smaller than what is already reported in the literature. The microscopic origin of the effect (either magnetization or ferric order) is relevant in a discussion of the details which is of interest for specialists: the physically measured effect is the MOKE or Faraday rotation, which is 10 times bigger in already studied materials than in Mn₃Sn.*

Reply: We thank the referee for pointing out that the time-resolved Faraday rotations bigger than 8 mrad were measured in FeBO₃. However, the Faraday rotation is measured in the transmission geometry and it is proportional to the sample thickness, so we think that the magnitude of the Faraday rotation cannot be directly compared with the surface sensitive Voigt and MOKE signals measured in the reflection geometry. In the referee-mentioned paper of PRL 89, 287401 (2002), the thickness of FeBO₃ sample used for Faraday rotation measurement is 300 μm, whereas the sample of Mn₃Sn we used for Voigt and MOKE measurements has thickness of only 40 nm. If the MOKE signal in Mn₃Sn is multiplied by the thickness ratio of 300/0.04, it would be much larger than the Faraday rotation measured in FeBO₃. The corresponding descriptions and reference [Ref. 28] have been included in the introduction (page 4) and discussion (page 7) of the revised manuscript.

Apart from the concern of the size of the MOKE effect, Mn₃Sn has attracted more interests just because of its special magnetic structure. The ferroic ordering of magnetic octupoles produces MOKE and AHE even in its fully compensated antiferromagnetic (AFM) state. For the inverse triangular spin structure in Mn₃Sn

with compensated spins, the time reversal symmetry is naturally broken, thus inducing linear MOKE signals. These unique properties have recently received great attention, and relevant studies on the static MOKE and AHE have been reported in Nature Photonics 12, 73 (2018) (cited by 99 times, Ref. 23) and Nature 527, 212 (2015) (cited by 434 times, Ref. 22), respectively. Therefore, Mn_3Sn with zero or vanishingly small magnetization is different with the class of antiferromagnets in insulators (like $FeBO_3$), which manifest weak ferromagnetism to yield the Faraday effect.

- *“These literatures report on the photonduced coherent magnon motion in two-sublattice antiferromagnets. Reviewer #3 calls those antiferromagnets as “noncollinear antiferramnagnets” just because the magnetic moments of two sublattice have a canting of $\sim 1^\circ$. but they essentially belong to the collinear antiferromagnets.”*

This is once again not true: collinear antiferromagnets have zero Faraday rotation, while $FeBO_3$ shows a huge Faraday rotation, as discussed above. So weak ferromagnets are NOT collinear antiferromagnets.

Reply: We agree with the referee that the nonzero Faraday rotation in two sublattice AFM is originated from the net magnetic moment resulted from the canted AFM spins. The previous reply of “they essentially belong to the collinear antiferromagnet” just means that the magnetic linear birefringence in these canted two-sublattice antiferromagnets is equivalent to that in the collinear antiferromagnet. But we acknowledge that this previous statement may cause misleading. The true meaning is that the magnetic linear birefringence used to detect the spin precession motion of canted two-sublattice antiferromagnets in the literature is still described by collinear AFM order L . This is an intrinsic difference with the three-sublattice antiferromagnet like Mn_3Sn , where the Voigt effect induced by the inverse triangular spin arrangement cannot be described by a collinear parameter L .

The novelty of the present work cannot be the theory, since the authors have already presented it in a recently published work (AIP Advances 11, 055003 (2021)). Furthermore the authors have already demonstrated that the Voigt effect can be employed to measure spin dynamics in antiferromagnets PRB 98, 134409 (2018). The arguments provided in my previous report and here above show that the same conclusions (i.e. ultrafast spin dynamics detected by Voigt effect and even Faraday effect) were demonstrated in the literature 10-20 years ago also for noncollinear systems, which are non collinear due to DMI, exactly like Mn_3Sn (as mentioned by the authors in the rebuttal).

Reply: We agree with the referee that our novelty is not the theory. We present the experimental results in this work as the novel findings and important advancements. The previous result we published in PRB 98, 134409 (2018) dealt with the collinear antiferromagnet CoO, different from the noncollinear antiferromagnet Mn₃Sn studied in this work. Although the ultrafast spin dynamics was detected by the Voigt and Faraday effect in the literature 10-20 years ago, it was detected mainly in the two-sublattice antiferromagnets with either collinear spins or slight canting of spins forming weak ferromagnetism. Even in the slightly canted antiferromagnet, the Voigt effect is described by an equivalent collinear antiferromagnetic order L . This is intrinsically different with the three-sublattice antiferromagnet like Mn₃Sn, where the Voigt effect induced by the inverse triangular spin arrangement cannot be described by a collinear parameter L .

I also would like to mention another very relevant paper: Nature Photonics 9, 25–29 (2014). In this work it is shown that both the Faraday effect and the Voigt effect can trace the ultrafast spin dynamics in YMnO₃, which is a noncollinear antiferromagnet with basically the same configuration as Mn₃Sn, i.e. three non collinear sublattices, in an hexagonal crystal structure with zero net magnetisation, as illustrated below (Fig. 1(b) and (c) from Nature Photonics 9, 25–29 (2014)):

Given this discussion, the authors cannot claim that “The effectiveness of Voigt effect in detecting AFM spin dynamics in antiferromagnets with noncollinear spin alignment is yet to be demonstrated.”, as they still do in the introductory part of the manuscript. This effect has already been demonstrated in a wide variety of materials, including samples almost identical if not identical to Mn₃Sn (Nature Photonics 9, 25–29 (2014)). Especially in view of Nature Photonics 9, 25–29 (2014), the only novel information of the manuscript is that also in Mn₃Sn this spin dynamics can be detected by means of the Voigt effect. The difference in the dynamics, i.e. coherent magnons VS demagnetisation has nothing to do with the magnetic structure: it is a consequence of optically pumping either a metal (free electrons induced demagnetisation) or an insulator with photon energy in the bandgap (light-scattering driven magnons). This point is reported in the review papers now included in the revised version of the paper.

Reply: We thank the referee to point out the paper of Nature Photonics 9, 25–29 (2014) (included as the new Ref. 20) reporting on the use of CME effect for detection of ultrafast spin dynamics in YMnO_3 , which is a noncollinear antiferromagnet with the triangular spin structure. In that paper, the CME effect is caused by the in-plane net magnetic moment when the spin precession motion is excited by laser pulse. The in-plane net magnetic moment breaks the isotropic magnetic symmetry in the Kagome plane. In contrast, in our work, the Voigt effect is caused by the intrinsic noncollinear Néel order with compensated spins, which breaks the magnetic symmetry between the in-plane and out-of-plane. In other words, the Voigt effect in our work is originated from the noncollinear Néel order and it is therefore used to detect the dynamics of the noncollinear Néel order of the compensated inverse triangular spins. To the best of our knowledge, the detection of the ultrafast dynamics of noncollinear Néel order has not been reported so far. In the work of YMnO_3 , the Voigt effect is originated from the oscillating net magnetic moment and thus cannot be used to detect its Néel order dynamics. In addition, the triangular spin structure in YMnO_3 does not yield linear MOKE signal. Nevertheless, considering the work in YMnO_3 and other few related works [Tzschaschel et al., Nature Communications **11**, 6142 (2020)] (Ref. 21), we realize that the previous statement “The effectiveness of Voigt effect in detecting AFM spin dynamics in antiferromagnets with noncollinear spin alignment is yet to be demonstrated.” is not accurate and it may cause misleading. We thus revised this sentence to “For such class of noncollinear antiferromagnets with compensated spin alignment, the effectiveness of Voigt effect produced by the inherent noncollinear AFM order in detecting its ultrafast Néel order dynamics is yet to be demonstrated”

We acknowledge that the difference in the dynamics, i.e. coherent magnons VS demagnetisation has nothing to do with the magnetic structure, but the coherent magnons may cause symmetry broken to induce oscillating Voigt signals or linear MOKE/Faraday signals, facilitating the detection, and on the contrary, demagnetisation maintains the magnetic symmetry and only modulates the inherent Voigt effect. If referring to the paper entitled “optical determination of Néel vector in a CuMnAs thin film antiferromagnet” published in Nature Photonics 11, 91 (2017) [Ref. 12] and considering that CuMnAs is merely a collinear antiferromagnet, one may get convinced that determination of the Néel order dynamics is a different and important issue as compared to the detection of the precessional AFM spins. Thus, we believe that demonstration of the effective separation of modulated MO Voigt effect related to the ultrafast demagnetisation (ultrafast quench of Néel order) in a noncollinear antiferromagnet is not marginal and incremental. We have included some descriptions of the work on ultrafast spins dynamics in YMnO_3 and two new references [Refs. 20-21] in the revised manuscript to clarify the novelty of our work.

Moreover, the direct comparison of ultrafast Voigt and MOKE effects in Mn_3Sn and understanding of ultrafast dynamics of the noncollinear Néel order are also important issues to be addressed since Mn_3Sn represents an important class of AFM materials. We hope that we have convinced the referee of our motivation and novelty.

Thus the novelty reported by the authors is incremental and marginal, one more material cane measured as a lot of materials already investigated. This is hardly suitable for the broader audience of Nature Communications. I stand by my original suggestion and evaluation of this work, finding it proper for Phys. Rev. B or New Journal of Physics.

Reply: Based on to the above replies, we hope that we have convinced the referee the clear motivation and novelty of our revised manuscript.

Reviewers' Comments:

Reviewer #3:

Remarks to the Author:

In my opinion the rebuttal to my previous report contains a technical mistake (described below) which, however, is not critical in the framework of the paper.

I also would like to note that some passages of the rebuttal were very hard to read. In particular, I couldn't understand the meaning sentences "If referring to the paper entitled....incremental." Nevertheless what is critical is the novelty factor and, once again, I see only a very marginal and incremental one in this work. In details:

- demagnetising a ferro- or antiferromagnetic metal when the material is heated is textbook physics.
- Ultrafast measurements of the MOKE and Voigt effects are abundant in the literature, also for non-collinear antiferromagnets. I agree with the authors on one point of their rebuttal: considering the specific material (Mn_3Sn), it may be true that a slightly different microscopic interpretation of the detected signal is required in comparison of the samples already investigated. However, the effect, i.e. MOKE and Voigt effects on ultrafast time scale in non-collinear antiferromagnets, cannot by any means be labeled as "novel". The details of the interpretations of an established phenomenon are of interest for the very specialised community investigated this compound and not for a broad audience.

For this reason I cannot recommend this work for publication in Nature Communications.

Technical issue

The authors write "the Faraday rotation is measured in the transmission geometry and it is proportional to the sample thickness, so we think that the magnitude of the Faraday rotation cannot be directly compared with the surface sensitive Voigt and MOKE signals measured in the reflection geometry." This is true for static Faraday rotation measurements. In the case of pump probe experiments the signal is not proportional to the thickness, since only the pump-induced effects are detected. The important lengths are the penetration depths of pump and probe, which are longer than the samples in PRL **89**, 287401 (2002), i.e. the entire detected volume is uniformly excited and probed.

In the case of the Mn_3Sn the penetration depth of both beams is a few nanometers long, since the material is a metal. So, even if a single crystal of 300 μm could be grown and employed in the experiments, the size of the signal would not change. Hence the intrinsic magnetic properties of this material generates magneto-optical signals 10 times smaller than the system discussed in PRL **89**, 287401 (2002)

Response letter (NCOMMS-21-03322B)

Dear Editor:

We thank Reviewer #3 again for giving further comments on our manuscript. Reviewer #3 still has concern of the novelty of our work, but he/she did not give any specific literature that may devaluate our new findings. We acknowledge that the Voigt effect has been employed to investigate the spin dynamics in various magnetic materials, but it has been limited in certain cases and the effectiveness of Voigt effect in detecting inherent noncollinear Néel order dynamics is yet to be demonstrated. In addition, the magnitude of the Voigt effect and the ultrafast dynamics of Néel order in compensated noncollinear antiferromagnets are also interesting and important topics to be explored. In this response letter, we have further addressed the concerns raised by reviewer #3. The manuscript has been revised accordingly and changes in the revised manuscript have been highlighted in yellow. We feel that the revised manuscript is now suitable for publication in Nature Communications.

In the following, we have included the point-to-point responses to the comments of the reviewer.

Point-to-point responses:

Reviewer #3:

In my opinion the rebuttal to my previous report contains a technical mistake (described below) which, however, is not critical in the framework of the paper. I also would like to note that some passages of the rebuttal were very hard to read. In particular, I couldn't understand the meaning sentences "If referring to the paper entitled....incremental."

Reply: In the previous response letter, we wrote: "We acknowledge that the difference in the dynamics, i.e. coherent magnons VS demagnetisation has nothing to do with the magnetic structure, but the coherent magnons may cause symmetry broken to induce oscillating Voigt signals or linear MOKE/Faraday signals, facilitating the detection, and on the contrary, demagnetisation maintains the magnetic symmetry and only modulates the inherent Voigt effect. If referring to the paper entitled "optical determination of Néel vector in a CuMnAs thin film antiferromagnet" published in Nature Photonics 11, 91 (2017) [Ref. 12] and considering that CuMnAs is merely a collinear antiferromagnet, one may get convinced that determination of the Néel order dynamics is a different and important issue as compared to the detection of the precessional AFM spins. Thus, we believe that demonstration of the effective separation of modulated MO Voigt effect related to the ultrafast demagnetisation

(ultrafast quench of Néel order) in a noncollinear antiferromagnet is not marginal and incremental.” Here, the sentence of “If referring to the paper entitled ...” should be revised as “When referring to the paper entitled...”. We are sorry for the grammar error. This passage just justifies that the detection of Néel order dynamics is a different story than the probe of precessional AFM spins. Though the Voigt effect had been used to detect the precessional AFM spins, the demonstration of detection of collinear Néel order was an important step forward and deserved the publication in prestigious journal of Nature Photonics. In our manuscript, we have further demonstrated the probe of non-collinear Néel order in a compensated antiferromagnet by employing the Voigt effect, which greatly broadens the application of the Voigt effect for study of the AFM spin dynamics in antiferromagnets.

Nevertheless what is critical is the novelty factor and, once again, I see only a very marginal and incremental one in this work. In details:

- *demagnetising a ferro- or antiferromagnetic metal when the material is heated is textbook physics.*

Reply: Although there are many works reporting on demagnetising ferro- and antiferromagnetic metals under ultrafast laser excitation, the underlying microscopic mechanism is still not very clear and remains under intensive debate. Hence, further study of ultrafast demagnetisation in metallic materials to clarify its physical mechanism is still a very important task. In our work, we have discovered that the demagnetisation of Mn_3Sn is quite unique because its demagnetisation time keeps nearly constant even when the temperature approaches the Néel transition temperature, in contrary to the common observations of slowing down demagnetisation near the magnetic transition temperature. We believe this invariant demagnetisation is due to the inverse triangular magnetic structure stabilized by the Dzyaloshinskii–Moriya interaction (DMI). This new finding distinguishes our work with previous work and it can deepen the understanding of demagnetisation process, in particular, the role of different magnetic energy and its modulation on the ultrafast demagnetisation. To further clarify the originality of our work, we have included a short passage and 5 new references in the introduction on page 4 (Ref. 31-35).

- Ultrafast measurements of the MOKE and Voigt effects are abundant in the literature, also for non-collinear antiferromagnets. I agree with the authors on one point of their rebuttal: considering the specific material (Mn_3Sn), it may be true that a slightly different microscopic interpretation of the detected signal is required in comparison of the samples already investigated. However, the effect, i.e. MOKE and Voigt effects on ultrafast time scale in noncollinear antiferromagnets, cannot by any

means be labeled as “novel”. The details of the interpretations of an established phenomenon are of interest for the very specialized community investigated this compound and not for a broad audience. For this reason I cannot recommend this work for publication in Nature Communications.

Reply: We acknowledge that the Voigt effect has been employed to investigate the spin dynamics in various AFM materials. But it has been limited in certain cases: 1) Collinearly spin-aligned AFM materials [Refs. 12-13], 2) Slightly spin-canted AFM materials [Refs. 14-19], and 3) Precession spin-induced symmetry breaking in non-collinear AFM materials [Refs. 20-21]. However, the effectiveness of Voigt effect in detecting inherent noncollinear Néel order dynamics is yet to be explored. In 2015, Satoh *et al.* reported on the transient Voigt effect caused by the net magnetization due to precessional spins in non-collinear AFM YMnO_3 , published in Nature Photonics **9**, 25 (2015) [Ref. 20]. In 2017, Saidl *et al.* demonstrated the determination of the ultrafast quench of Néel vector in a collinear AFM CuMnAs thin film using the Voigt effect, published in Nature Photonics **11**, 91 (2017) [Ref. 12]. We believe that our work of demonstration of ultrafast dynamics of Néel order in a noncollinear AFM Mn_3Sn is an important step forward. We did not intend to label the utilization of Voigt effect in detecting spin dynamics of AFM materials as a novel technique, but emphasize that we have extended this technique to study the inherent non-collinear Néel order in an important class of AFM materials. In order to avoid the misunderstanding of our meaning, we have changed some phrases in the main text, deleting the word “new” or “novel” when describing the Voigt effect for the probe of the noncollinear AFM spin dynamics.

Technical issue

The authors write “the Faraday rotation is measured in the transmission geometry and it is proportional to the sample thickness, so we think that the magnitude of the Faraday rotation cannot be directly compared with the surface sensitive Voigt and MOKE signals measured in the reflection geometry.” This is true for static Faraday rotation measurements. In the case of pump probe experiments the signal is not proportional to the thickness, since only the pump-induced effects are detected. The important lengths are the penetration depths of pump and probe, which are longer than the samples in PRL **89**, 287401 (2002), i.e. the entire detected volume is uniformly excited and probed.

In the case of the Mn_3Sn the penetration depth of both beams is a few nanometers long, since the material is a metal. So, even if a single crystal of $300\ \mu\text{m}$ could be grown and employed in the experiments, the size of the signal would not change. Hence the intrinsic magnetic properties of this material generates magneto-optical

signals 10 times smaller than the system discussed in PRL **89**, 287401 (2002).

Reply: Thanks for pointing out that the penetration depth of our pump and probe beams for the Mn_3Sn is at the scale of 10 nanometers, so the single crystal of 300- μm cannot be used for the transmission measurement. The fact that the large Voigt and MOKE signals in Mn_3Sn are from a few nanometers just proves the remarkable linear and quadratic magneto-optical responses in this material. In other words, when comparing the MO responses of nanometer-scale thin films, Mn_3Sn exhibits much larger MO signals compared to the FeBO_3 discussed in PRL **89**, 287401 (2002). Moreover, as mentioned in the PRL paper, the amplitude of the Faraday signal peak is a linear function of the pump fluence, and its Faraday rotation was measured at the pump fluence of $\sim 30 \text{ mJ/cm}^2$. As a contrast, in our work, the pump fluence is only about 0.75 mJ/cm^2 . Accordingly, we have revised the text on page 7.

With kind regards,

Haibin Zhao

Ph.D., Professor

Dep. of Opt. Sci. & Eng.

Fudan Univ., Shanghai, China